# Benchmarking Egocentric Multimodal Goal Inference for Assistive Wearable Agents

**Vijay Veerabadran[1] Fanyi Xiao[2] Nitin Kamra[1] Pedro Matias[1]**
**Joy Chen[2] Caley Drooff[1] Brett D Roads[1]\* Riley Williams[1]\* Ethan Henderson[1] Xuanyi Zhao[2]**
**Kevin Carlberg[1]\* Joseph Tighe[2] Karl Ridgeway[1]**
[1] Meta Reality Labs [2] Meta FAIR
{vveerabadran@meta.com, kridgeway@meta.com}
Project website: https://facebookresearch.github.io/WAGIBench/

## Abstract

There has been a surge of interest in assistive wearable agents: agents embodied in wearable form factors (e.g., smart glasses) who take assistive actions toward a user's goal/query (e.g. "Where did I leave my keys?"). In this work, we consider the important complementary problem of inferring that goal from multi-modal contextual observations. Solving this "goal inference" problem holds the promise of eliminating the effort needed to interact with such an agent. This work focuses on creating **WAGIBench**, a strong benchmark to measure progress in solving this problem using vision-language models (VLMs). Given the limited prior work in this area, we collected a novel dataset comprising 29 hours of multimodal data from 348 participants across 3,477 recordings, featuring ground-truth goals alongside accompanying visual, audio, digital, and longitudinal contextual observations. We validate that human performance exceeds model performance, achieving 93% multiple-choice accuracy compared with 84% for the best-performing VLM. Generative benchmark results that evaluate several families of modern vision-language models show that larger models perform significantly better on the task, yet remain far from practical usefulness, as they produce relevant goals only 55% of the time. Through a modality ablation, we show that models benefit from extra information in relevant modalities with minimal performance degradation from irrelevant modalities.

## 1 Introduction

An *assistive wearable agent* (or wearable agent) is an AI agent that observes the world from a user-centric perspective and takes actions to achieve a user-provided query. There have been a number of recent works on assistive wearable agents including digital agents in mobile phones [17] or web-browsers [9, 12], superhuman memory agents [20, 21], and assistance for the visually impaired [19] but a challenge for all of these agents is the need to fully specify a query or have long interactions before the agent understands the user's goal.

To address this, we propose a *goal-inference* module that infers useful goals for the wearable agent to execute, eliminating or greatly reducing the length of the user query required. An ideal goal inference module observes all possible passive behavioral cues across various modalities, including egocentric vision, egocentric audio, and digital context (e.g., calendar state, search history, notes, etc) and needs to maintain a *longitudinal* history of these cues so that it can personalize its prediction based on the user's past actions and preferences. We model this problem as a multi-modal language task, with *video*, *audio*, *digital*, and *longitudinal* context as input. Figure 1 visualizes how these 4 modalities

---

\*Work done while at Meta

39th Conference on Neural Information Processing Systems (NeurIPS 2025) Track on Datasets and Benchmarks.

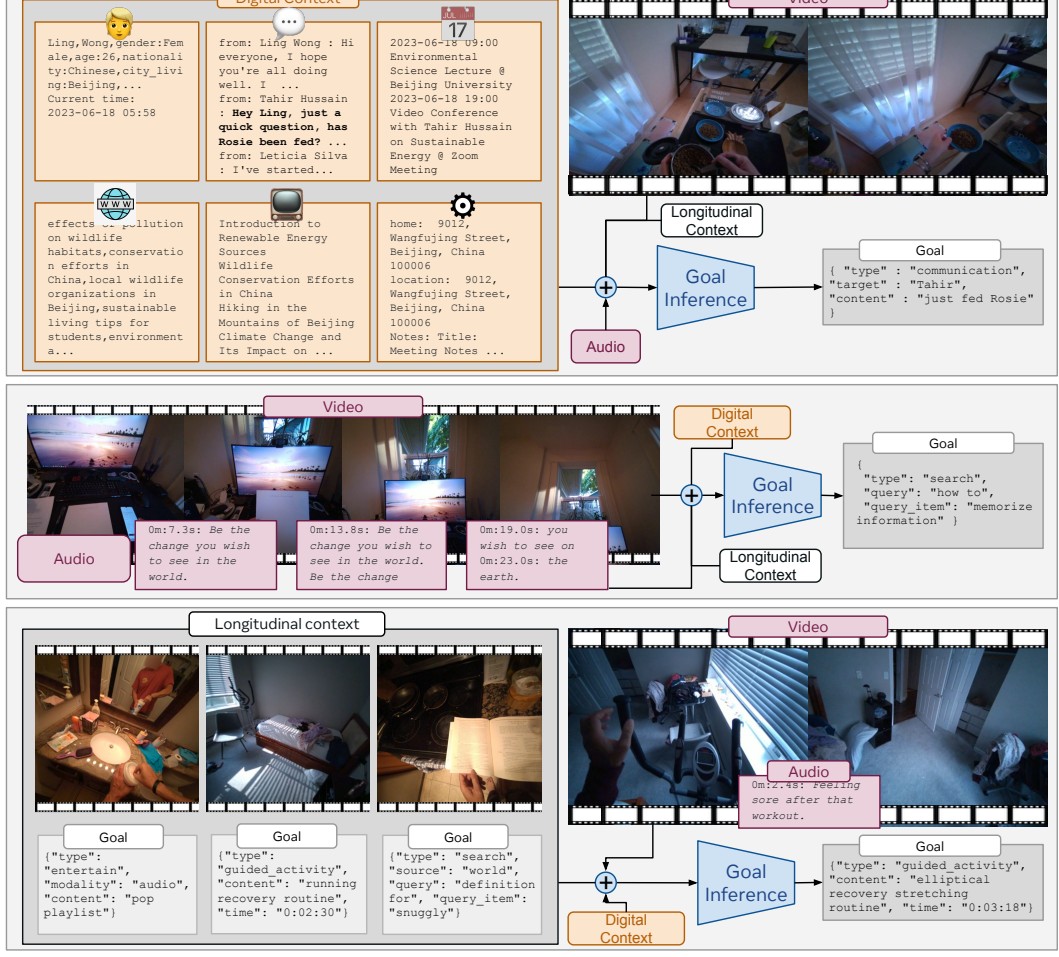

Figure 1: Three multi-modal samples from the benchmark. In the top row, the video and digital contexts are relevant to the prediction problem, and audio/longitudinal are noise (we call this the $S_{\text{VD}}$ subset). In the middle row, video and audio are relevant ($S_{\text{VA}}$). In the bottom row, the video, audio, and longitudinal contexts are relevant ($S_{\text{VAL}}$).

are used for different user scenarios. This work aims to develop a robust and relevant benchmark for this new domain of assistive wearable agents.

Benchmarking goal inference for wearable agents is challenging due to the lack of ecologically valid datasets with accurate "ground truth" goals. Existing datasets like Ego4D [11] are often re-annotated using LLMs [1, 4, 6, 7, 14, 20], but they under-represent key moments of utility for wearable agents (e.g., user leaves their house without their keys), and lack a source of ground truth for goals. We address these issues by introducing WAGIBench, a novel multimodal egocentric goal inference benchmark for wearable agents. For designing WAGIBench, we collected a novel dataset (Figure 1) through scripted interactions, covering various digital apps and environments to ensure the correctness of reference goals, which we quantified *post hoc* by humans ability to perform the goal-inference task. Table 2 systematically compares our dataset to other egocentric datasets.

To properly measure the effectiveness of a goal-inference system we not only need to have *video*, *audio*, *digital*, and *longitudinal* modalities present in the dataset, we need to ensure that each modality is required on at least a sub-set of the scenarios. To this end, we not only include all 4 modalities in each recording as shown in Figure 1, but design our scenarios with the required contextual modalities in mind and validate the relevance of the modalities through an ablation study. Just as important as data curation, the methodology for calculating the evaluation metric is critical to an effective benchmark. In this work we leverage two canonical paradigms: *discriminative evaluation*,

Table 1: Comparison of wearable assistant benchmarks. For modalities, 🎥 indicates egocentric vision, 🎤 indicates egocentric audio, 🖥 indicates digital context, and 🗒 indicates human-generated narrations. $(\cdot) \times T$ indicates the benchmark is *longitudinal*, where the model needs to process long time sequences or multiple episodes in order to succeed.

| Paper | Dataset | Videos | Questions | Ground Truth | Task | Modalities |
|-------|---------|--------|-----------|--------------|------|------------|
| MM-Ego | Ego4D | 629 | 7,026 | LLM (narrations) | Agent Policy | 🎥 |
| EgoLife | EgoLife | 6 | 6,000 | LLM (captions) | Agent Policy | (🎥, 🎤) $\times$ T |
| PARSE-Ego4D | Ego4D | 10,133 | 19,255 | LLM (narrations) | Goal Inference | 🎥 or 🗒 |
| Ours | Ours | 3,477 | 3,477 | Scripted | Goal Inference | (🎥, 🎤, 🖥) $\times$ T |

Table 2: (left) Comparison of Egocentric datasets. (right) Per-modality stats of our dataset.

| Dataset | Domain | Particip. | Modal. | Videos | Hrs. | Longit. |
|---------|--------|-----------|--------|--------|------|---------|
| Ego4D | Various | 923 | 🎥🎤 | 9,600 | 3,670 | No |
| EgoLife | Home/Outdoor | 6 | 🎥🎤 | 6 | 266 | Yes |
| LEMMA | Home | 8 | 🎥🎤 | 324 | 10.1 | No |
| EPIC-Kitchens | Kitchen | 37 | 🎥🎤 | 700 | 100 | No |
| EGTEA Gaze+ | Kitchen | 32 | 🎥🎤 | 86 | 28 | No |
| Ours | Various | 348 | 🎥🎤🖥 | 3,477 | 29 | Yes |

| | Vision | Audio/Speech | Digital | Longitudinal |
|---|--------|--------------|---------|--------------|
| total | 29.3h | 6.9h
57.6k words | 29.6MB
7 apps | 12.2MB
142.8h |
| avg(std)
per sample | 30.4(27.1)s | 7.2(14.0)s
16.6(35.3) words | 8.7(1.0)KB | 147.8(119.1)s
3.6(2.9)KB |

which is implemented via multiple choice questions (MCQs), and *generative evaluation*, which is implemented via an LLM judge model. We perform a meta-evaluation comparing the paradigms to the gold standard of human evaluation of the generative performance, and find that LLM Judge is superior to MCQ, and even on par with human raters.

In summary we propose WAGIBench, a benchmark to measure the performance of goal inference for *assistive wearable agents*. Our benchmark provides (1) a novel scripted dataset for egocentric goal inference with 348 participants generating 3,477 video clips (29 hours), each with a digital context and reference goal; (2) the first benchmark incorporating *video*, *audio*, *digital*, and *longitudinal* context; and (3) an LLM judge using reference or scripted cues to substitute human judges without accuracy loss.

## 2 Dataset

Our dataset comprises 3,477 (observation, goal)-pairs, each featuring an egocentric view with four modalities: vision, audio, digital, and longitudinal. To ensure diversity and high annotation quality, we collected observations from a large pool of 348 participants, limiting each to a fixed set of 165 scripted scenarios covering various themes and parameterizable goal classes. The participants had diverse cultural backgrounds, but were predominantly recruited from the state of Washington, USA, which may somewhat reduce diversity. We would like to emphasize however, that scripting participant actions reduces potential biases caused by patterns of behavior related to their demographics. Goal inference in indoor environments is still an unsolved, challenging problem, and our scenarios cover a large proportion of such indoor themes mixed with a few outdoor themes which are more challenging to collect from a logistics standpoint. Each pair is annotated with relevant modalities for goal inference. Vision dominates, while other modalities are well-represented individually or in combination (Figure 2). Table 2 (right) provides a statistical summary of each modality, highlighting dataset size and variance. We also analyzed environmental diversity using the vision modality, accounting for 10 common location settings in mixed lighting conditions (Figure 2, right)[2]. On average, each script was recorded ~21 times among ~6 participants, with each participant recording ~10 videos for ~8 scripts. The remainder of this section details the curation of scripts (Section 2.1), as well as the collection methodology of each modality (Sections 2.2 through 2.4).

---

[2]We used Qwen2.5-VL-72B zero-shot on 8 evenly sampled frames to classify each video location and lighting conditions.

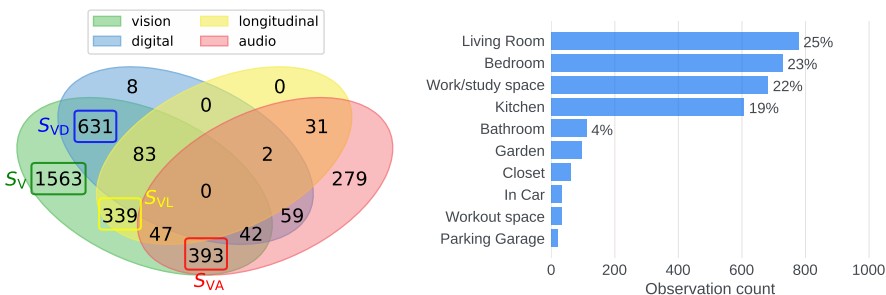

Figure 2: (left) Venn diagram showing the spread of recordings where different combinations of modalities are relevant to the goal-inference task (all modalities are always available). Subsets used in our experiments are tagged with the name we use to refer to them ($S_V$, $S_{VA}$, $S_{VD}$, $S_{VL}$). (right) Histogram of recording locations as estimated by an automated VLM classification.

## 2.1 Script Generation

Ideally, our scripts should be *ecological valid*, including common situations where assistive wearable agents can be useful, they should be *diverse* in terms of the environments, situations, and wearable agent goals, and should be *multi-modal*, with situations involving various combinations of modalities. To develop ecologically valid and environmentally diverse scenarios for wearable agents, we began with popular apps from mobile app stores and common environments like Kitchen, Living Room, Office, Transit, Outdoors, Bedroom, Gym, and Social gatherings. Script ideas were generated to cover these apps in various settings, and Llama 3.2 was used to finalize scripts for data collection. An example script usually involves some *prework* e.g., "Have recyclables that you can sort." and *instructions* such as "1. Put an object in the recycling bin; 2. Have an additional <item> of a different type you would like to recycle in front of you". Each script has a "reference" (ground truth) goal like "search whether <item> can be recycled". To promote goal diversity, scripts include *variables* that vary between participants, such as the "<item>" that the participant gets to choose based on their preferences. Each script's digital goal was categorized into a structured representation with an app type (e.g., web search, shopping, memory storage, timers/reminders, guided activities, messaging, smart lights, translation, entertainment) and app-specific arguments. Web search was the most common app type, reflecting frequent use cases like recipe searches and product reviews.

To ensure multi-modality, we designed scripts where various combinations of modalities are relevant for goal inference. Cues were categorized by relevant modality (V=vision, A=audio, D=digital, L=longitudinal) and described textually. Visual cue example: "#c looks at empty box of <ingredient>", audio cue: "#c sings to warm up their voice", digital cue: "#c checks calendar for 'travel to <destination>' ", and longitudinal cue: "#c likes to meditate in bed" (#c indicates the camera wearer). After authoring, scripts were sent to participants for audiovisual context recording. To ensure privacy, digital states were synthesized using LLMs conditioned on digital context cues.

## 2.2 Audiovisual Context Generation

Each script was recorded by an average of 6 participants using Meta Aria glasses [10], capturing egocentric video and audio. Participants recorded each script multiple times on different days, allowing for naturalistic variation and longitudinal evaluation, with an average of 20 recordings per script across participants. Post-recording, three raters annotated the videos, each providing: (1) a *quality* score $\in \{\text{accept}, \text{reject}\}$, (2) *variable annotations* as key-value pairs (e.g., {item: styrofoam}, {item: toilet paper}), and (3) a *context window* with start- and end-times to exclude irrelevant video portions. The quality score was based on adherence to the script, data quality, and ecological validity. A recording was accepted if: (a) at least two raters assigned a quality score of accept, (b) variable annotations agreement exceeded 0.5, and (c) the context windows' average pairwise intersection-over-union was above 0.7. Variable agreement was calculated using the smallest pairwise cosine similarities among raters' annotations, determined by sentenceBERT [18]. Recordings rejected by at least one rater were re-evaluated by researchers. This process retained approximately 80% of participant recordings.

## 2.3 Digital Context Generation

We designed a pipeline for using large language models to generate rich digital contexts representing the internal app states of seven widely-used apps: *Calendar*, *Messaging*, *Notes*, *Search*, *Videos*, *Maps*, and *Music*. Forty-four scripts have at least one relevant *digital cue* (e.g., "Tahir just sent a message asking if Rosie has been fed"). The digital cue text is used to condition the generation of the associated app. Other app states are generated without conditioning. An example abbreviated generation is shown in Figure 1 (top). In total, 856 observation-goal pairs have some relevant digital context, and the remaining 2,621 observations only have irrelevant digital context. The generation proceeds by first sampling a *global state*, consisting of a set of six *personas*, each with a name, gender, nationality, occupation, etc, one of which is tagged as the main or egocentric persona, and the current date/time. Each of the seven app states is generated conditioned on both the global state and on the relevant digital context cue if available. Table 2 shows data sizes of digital context. Note that that digital context contains a lot of non-relevant information, which models must ignore to correctly infer the user's goals. A complete description of the generation process can be found in the Appendix.

## 2.4 Longitudinal Context Generation

We take inspiration from Zhang et al. [23], where longitudinal histories are synthesized by using annotations to concatenate different source clips. However, in that work the videos are drawn from Ego4D and can span different users with in different locations and environments. Our histories combine recordings only from the same user in the same environment. Figure 1 (bottom) shows an example generated longitudinal history, where each video represents an episode of interaction with the wearable agent and a goal provided by the user. The model needs to see a previous use of an app to guide the user through a recovery/stretching routine to know that it's their habit to use this app to recover after vigorous workouts. However in order to predict the current goal correctly, the model also needs to recognize that the workout has changed from running to using the elliptical machine (from video context). Per our data collection paradigm, each participant performed multiple repeats of each assigned script, on different days, and with naturalistic variation between repeats. Some scripts are "longitudinal", indicating that it would make sense for a user to repeat the scenario, e.g., user adds an item they are out of to their grocery list. An example non-longitudinal script could be learning how to set up a tent (typically not done more than once). Observations recorded with longitudinal cues make up the longitudinal set (yellow ellipse in Figure 2), and 25 of the 165 total scripts (535 of 3,477 observations) are longitudinal.

**History Bank Generation.** For each observation-goal pair, we create a longitudinal history bank of previous audiovisual observations, each one represented as a textual caption of the video generated by a VLM combined with the audio transcript. The history bank is populated with 5 *support* observations from a participant (in some cases fewer if a participant did not complete their allocated recordings). If the observation is $\in L$, then one history bank observation is a *positive support* (shares a script with the observation), otherwise they are all *negative supports* (dissimilar scripts). During benchmark evaluation, the history bank is shuffled. In principle, history bank context can be represented as raw video/audio observations. To achieve realistic prompt lengths with current VLMs, we represent them with *Socratic context*[22]: detailed captions of videos in the longitudinal history bank using VLMs and these text captions are used to represent longitudinal cues during evaluation. Two VLMs (`Qwen2.5-72B` and `InternVL-78B`) generate detailed captions summarizing the events in an input longitudinal video, which are then checked for inter-caption consistency, and a summary of events that co-occur in both captions is used as Socratic context for the longitudinal video. Models are provided with the Socratic text of the entire history bank.

## 3 Benchmark Tasks

### 3.1 Discriminative Evaluation: Multiple Choice Questions

Multiple Choice Questions (MCQs) are highly interpretable but exhibit significant bias compared to the gold standard of human evaluation of generated goals by limiting choices to a fixed-set of options. Core to the design of MCQ is the selection of challenging distractors without inadvertently choosing positives. Approaches include generating distractors using LLMs [13, 14, 21], having annotators create them [20], or sampling from the dataset [4]. We utilize MCQs with dataset-sampled distractors as one of our evaluation methods. Our method evaluates performance of inferring a goal and its finegrained parameters; this is fundamentally different from the method in [1] which studies coarse-grained discriminative evaluation at the goal level (ignoring parameters).

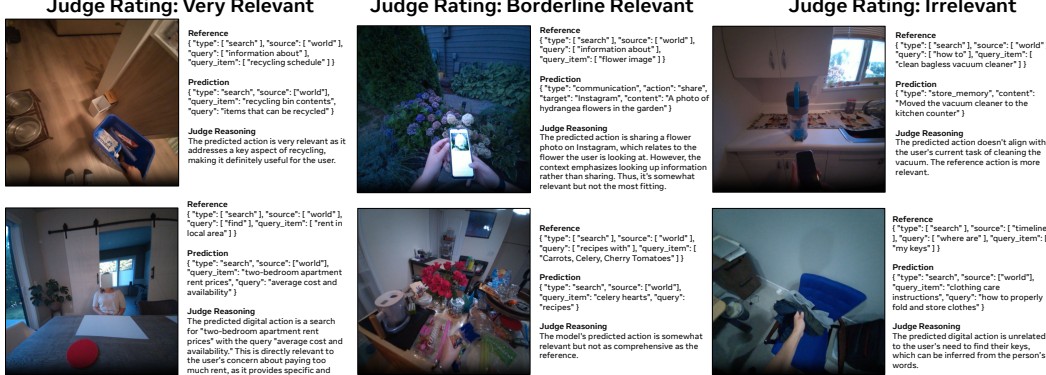

Figure 3: Example LLM Judge responses for goal inference predictions, along with reference goals and the judge's reasoning trace. Best viewed when zoomed in.

A multiple choice question problem is composed of an observation (vision, audio, digital, and longitudinal context), a reference goal, and a set of N distractor goals (we set N=3). We generate two distinct sets of distractors to focus on coarse- and fine-grained problems, respectively: A set of *similar* distractors for "search how to train a dog to sit" might be { search how to take care of a tree, search for uplifting videos such a dog videos, search for how to memorize information}, and a set of *dissimilar* distractors could be { search for fact check: did henry cavill star in man of steel?, play a guided meditation for moning, store memory: add chicken tenderloins to grocery list }. To sample distractors in an option-set, we first map goals onto embeddings by converting them into natural language (e.g., "Do a search for how to revive a dying evergreen") and processing them with the `paraphrase-mpnet-base-v2` sentenceBERT model [18]. For each reference goal, we compute its cosine similarity to the entire set of other reference goals in the dataset. *Similar* distractors are sampled from between the 95th and 99th percentile of similar goals, and *dissimilar* distractors are sampled from between the 0th and 80th percentile. To ensure a diverse set of distractors $D$ for a single MCQ, we adopt a greedy sampling strategy:

$$D_i = \begin{cases} \texttt{sample}\big(U(C)\big) & \text{if } i = 1 \\ \arg\min_{c \in C \setminus D_{<i}} \big( \max_{j=1}^{i-1} \texttt{sim}(D_j, c) \big) & \text{if } 1 < i < 4, \end{cases} \tag{1}$$

where $C$ is the set of candidate distractors in the target percentile range, $U$ is a uniform distribution, and `sim` gives the embedding cosine similarity of two goals. For each sample in the dataset we generate 1 "similar" MCQ and 1 "dissimilar" MCQ, totaling ~7k MCQs. In the prompt passed to the multimodal-language model, we shuffle the option-set and prepend the letters A, B, C, or D to indicate the index of the option, and ask the model to produce the correct index.

## 3.2 Generative Evaluation: LLM Judge

Goal inference is ultimately not a fixed-set task: the model needs to generate the goal in an open-set fashion. An approach to open-set evaluation involves scoring responses using an LLM Judge model, which can be prompted to compare a generated and a reference output [6, 7]. Other open-set/reference-based metrics include text- or embedding-based similarity measures like RougeL or BertScore [23], and negative log-likelihood of the reference [1], which tends to overemphasize long strings and does not consider negatives. For the goal-inference problem, we want the judge to be flexible in its interpretation of a predicted goal, as there may be multiple relevant goals for an observation. We therefore adopt an LLM-as-judge model[3] (example outputs shown in Figure 3) that is parameterized with the predicted goal, a reference goal, and the cues from the script. We experiment with variations in Section 4.3. The LLM Judge is tasked to output a score that's either 1.0 for "very relevant", 0.5 for "borderline relevant" or 0 for "irrelevant". In addition, we also ask the LLM Judge to output any reasoning leading to the score for easy interpretation. See Figure 5 (left) for a visualization of the LLM Judge's working.

---

[3]We use DeepSeek-R1-Distill-Llama-70B [8]

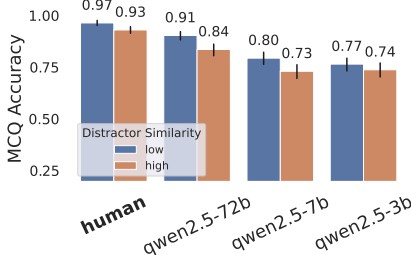

| Model | Size | MCQ | Generative |
|-------|------|------|------------|
| Llama3.2 | 11B | 0.4311 | 0.3197 |
| InternVL-2B | 2B | 0.4422 | 0.2134 |
| InternVL-8B | 8B | 0.6741 | 0.3503 |
| InternVL-78B | 78B | 0.8680 | 0.4866 |
| Qwen-3B | 3B | 0.7153 | 0.2468 |
| Qwen-7B | 7B | 0.7754 | 0.3999 |
| Qwen-72B | 72B | 0.8755 | 0.4980 |
| GPT-4.1 | - | **0.8774** | **0.5498** |

Figure 4: (left) Human study subset MCQ accuracy results for humans and the Qwen models, ordered by the mean accuracy of each model across distractor similarities. Error bars indicate bootstrapped 95% confidence intervals of the mean. (right) Mean performance on full dataset with all modality inputs for MCQ and Generative tasks.

## 4 Experiments

### 4.1 Evaluation Methodology

We choose to evaluate the following three representative and performant model families on our benchmark: Llama models [15], Qwen2.5-VL models [2], and InternVL-2.5 models [5], for their diverse model architectures and training recipes. For each model family, we evaluate several model variants with different sizes. Specifically, we include `Llama-3.2-11B-Vision` for Llama models, `Qwen2.5-VL-3B/7B/72B` for Qwen2.5-VL models, and `InternVL2.5-MPO-2B/8B/78B` for InternVL-2.5 models. In addition, we also include GPT-4.1 as the leading closed-source model. Since none of these models can process audio natively, we pre-process the audio using an internal speaker diarization toolkit and transcribe them using Whisper [16]. Further details can be found in the appendix.

**Human study subset.** To run our two human studies (human discriminative predictability and the meta-evaluation), we extracted a high inter-annotator agreement subset of 586 samples from our dataset (described in Section 2). We used the following criteria: (1) sample's relevant modalities were limited to audio and/or vision (humans had a hard time parsing the large and complex digital and longitudinal observations using our web-based annotation tools), (2) sample's context window annotation had an average pairwise intersection-over-union of above 0.95, (3) sample's *variable annotations agreement scores* of above 0.99 (see Section 2.2 for details on this score).

### 4.2 Performance on Discriminative Evaluation

To quantify the predictability of goals in our dataset, we established a human baseline performance for goal inference. We designed a web-based annotation tool that presented a rater with a playable video (including audio), a set of MCQ options, and a prompt to choose the best goal for the camera-wearer. We had a total of 584 MCQ's (we used two for training), where each question was answered by 3 raters out of a pool of 11 total raters (who had no experience with the dataset before). The results of the predictability study are shown in Figure 4. We find that human accuracy sets an upper bound on model performance: accuracy on the dissimilar MCQ options is close to saturating at 97% (91% for the next best model), and accuracy on similar MCQ's is slightly lower at 93%, with 84% for the best model (we note human inter-rater agreement is quite high, at 90.8% according to the choice-consistency metric proposed in Li et al. [13]).

### 4.3 Choosing the Best Judge: Generative Task Meta-Evaluation

Next, we assess different automatic evaluation techniques on the generative task. Techniques are compared to a "gold standard" evaluation of human raters scoring goals simply by watching the video while have no access to reference or cues that were part of the script.

We evaluate the generated goals with the LLM Judge as shown in Figure 5, and experiment with different judge inputs to augment the prediction. Two variants with no reference-bias: 1) socratic, 2) cues, and three variants with reference-bias (assessing the judge's reliance on an annotated reference goal): 3) reference, 4) socratic+reference, 5) cues + reference. "Reference" refers to feeding the

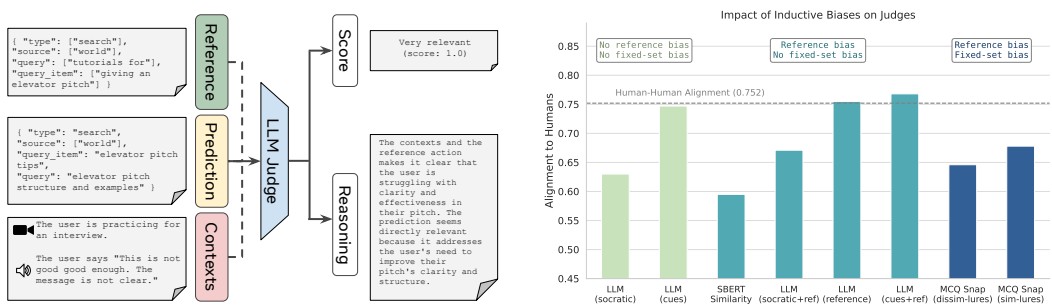

Figure 5: (left) LLM-as-Judge for Generative Evaluation. (right) Alignment between Human raters and Judges with different inductive biases.

reference goal, "Socratic" refers to feeding a description of the video[4], and "cues" refers to feeding cue descriptions from the script. To assess the impact of fixed-set bias of MCQ on the generative mode, we also consider a setting called "Snap-MCQ" where a generated output is mapped onto a fixed set of MCQ options by choosing the most similar one according to sentenceBERT [18], and outputting a rating of 1 if the chosen option matches the reference goal or 0 otherwise. Meanwhile, to assess the impact of an LLM's world knowledge on rating goals, we also add a baseline (denoted as "SBERT Similarity") where instead of using LLMs, we simply compute the similarity between the generated and reference goals using sentenceBERT representation. The raters reviewed 7 goals per video, each generated by different VLMs, in the high quality subset.

Human raters watched the video and assigned a score to each video-goal pair: {1: very relevant, 0.5: borderline relevant, 0: irrelevant}. To reduce the effect of different calibrations between human raters and judge models, we evaluate human-model agreement according to the *pairwise comparison accuracy*. This metric determines how often a judge model agrees with a human rater when comparing pairs of predictions ($<$, $=$, or $>$) based on their corresponding relevance scores (for all prediction pairs generated by different models for the same observation). We show the results from this analysis in Figure 5. In general, we found it being a helpful inductive bias to enable access to the reference goal for the judges. In particular, we find that the LLM Judge model parameterized with both reference and script cues performs best, with 76.8% alignment, and is indistinguishable from human-human agreement (75.2%). Both "Snap-MCQ" and "SBERT Similarity" perform significantly worse at 67.8% and 59.5%, respectively. Socratic context also underperforms at 63.0%.

## 4.4 Model Performance and Modality Ablation

In this section, we discuss the results of evaluating a suite of VLMs of varying model sizes on the full benchmark. We group the models mentioned in Section 4.1 into small (# params $\leq$ 3B), medium (# params $\in$ [7B, 17B] ) and large (# params $\geq$ 72B) parameter variants to study the effect of model size on performance.

**Performance is correlated with model size.** We evaluated the above models on discriminative and generative benchmarks on the full set of videos, we show these results in Fig. 4. We observe a positive correlation between model size and performance on both tasks, i.e., within each model family, the larger model size outperforms the smaller model size.

**Effect of input context modality.** We represent the observed context with the help of 4 modalities. For simplicity, we indicate combinations of modalities with a simple string, e.g., *context-modality =* V for vision-only context, VA for vision and audio input, etc. In the next suite of experiments shown in Fig. 6, we assessed the relative importance of each modality by testing different combinations of input modality context on subsets of our data. For example, to test the importance of audio information, we tested whether context prompts with the audio transcription (*context-modality =* VA), outperformed a context prompt without audio context (*context-modality =* V) on the subset of data where both vision and audio are relevant, $S_{VA}$ (Figure 6, left). We performed this evaluation for each of the three non-visual modalities and observe that multi-modal context significantly enhances performance over unimodal vision-only context. The respective modality-specific gains are as large as 35% on the MCQ task and 30% on the generative evaluation.

---

[4]Generated with the same captioning method used for longitudinal context

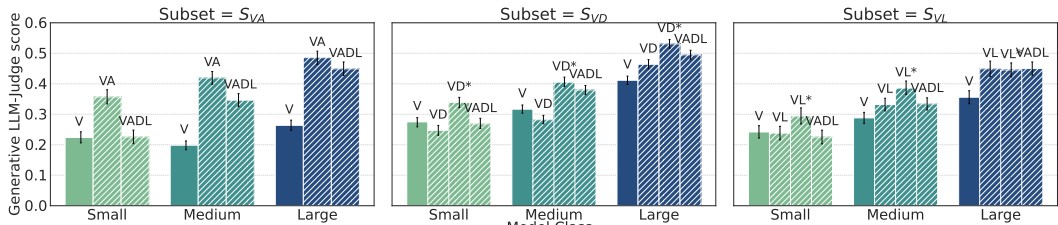

Figure 6: Ablating context modality on generative evaluation. The subplots show ablation results on three disjoint subsets of our data, requiring specific task-relevant modalities: $S_{VA}$ (vision-audio: left), $S_{VD}$ (vision-digital: middle), and $S_{VL}$ (vision-longitudinal: right). Each subplot groups the models into three classes (small, medium and large) based on their sizes and evaluates them with a different combination of input modalities (captioned on each bar) relevant for that subset. Error bars represent 95% bootstrapped confidence intervals of the mean.

**Signal-to-noise ratio in digital and longitudinal modalities.** Compared to audio, we observed less benefit from adding digital or longitudinal context, and hypothesize that this is due to their low signal-to-noise ratio. To verify this, we created two synthetic "high-signal" modalities, $D^*$ and $L^*$. For the digital modality $D^*$, we included only the relevant app sub-states generated directly from digital context cues. Similarly for the longitudinal modality $L^*$, we used only positive-support context. In Fig. 6, we can see that for $S_{VD}$ models using the *context-modality* = VD$^*$ outperform (as much as 12% gain) models using the VD modality (similarly, VL$^*$ outperforms VL by at most 5.6% on $S_{VL}$). For large models, the gap in performance for high-signal modalities shrinks, i.e., larger models are better able to filter out noise from these modalities.

**Effect of using all context modalities.** We evaluated the performance of using all modalities, i.e., *context-modality* = VADL. We observe that the large models are able to disentangle the relevant features among a mix of modalities with both task-relevant and distracting features, but small/medium models see interference. We refer to Fig. 6 for the above mentioned performance comparisons.

### 4.5 Visualization of Generated Goals

Figure 3 shows predicted goals and LLM Judge responses for them. Specifically, we run the Qwen2.5-VL-72B-Instruct model with vision+audio (VA) inputs to predict the goals. We group predicted goals into three buckets by their LLM Judge scores. The top-left example shows a prediction that is different to the reference but still considered very relevant by the judge, whereas in the bottom-left example the prediction goes into a finer granularity than the reference (i.e., "two-bedroom rent price" vs "rent in local area"). The top-middle example is a scenario that requires nuanced understanding to differentiate between "search info about the flower" vs "share the flower image", whereas the bottom-middle one shows a case where the prediction is relevant but not comprehensive (i.e. incomplete ingredients for recipe). For the top-right goal, the model falls back to "store_memory" action as it has trouble understanding the user's intention. For the bottom-right goal, failure is due to speech transcription error (i.e., person says he's looking for his keys, but the transcription is empty).

## 5 Conclusion and Discussion

**Novel Dataset and Benchmark.** We introduced WAGIBench: the first dataset for wearable agent goal inference across diverse locations, situations, and users, encompassing vision, audio, digital, and longitudinal modalities. Human studies show a 93% accuracy in multiple-choice goal prediction, with large VLMs achieving 83%. For the generative task, a meta-evaluation revealed that an LLM judge, conditioned on a reference goal or script cues, can effectively substitute human evaluators. We found the fixed-set bias in MCQ and noise in Socratic video descriptions reduce alignment with human preferences.

**Challenges for Future VLMs.** In the generative setting, state-of-the-art models predict relevant goals only 55% of the time, indicating room for improvement. A strong correlation between model size and performance suggests future VLMs may bring performance to a usable level. We see even bigger performance gaps on small models which are required for efficient inference on wearable/edge devices. Our modality ablation studies show VLMs benefit from all modalities but struggle with

digital and longitudinal due to low signal-to-noise ratios, whereas audio shows more benefit due to the strong inductive bias of automatic speech recognition.

**Limitations.** Human discriminative predictability was validated only for vision and audio modalities, limited by the complexity of digital and longitudinal contexts. WAGIBench assumes the user initiates all interactions, but a proactive system inferring both *when* and *what* actions to take would require a new dataset with more negative samples. Also, longitudinal history can capture more than routine behaviors, such as relevant world states (whether the users home is clean) and user preferences (is the user vegetarian). In future work, we aim to expand the dataset to include diverse longitudinal cues.

**Broader Impact.** Reducing interaction friction with assistive wearable agents via goal inference could improve accessibility for individuals with disabilities and enhance user experience for the wider population. Reliance on large-scale datasets and computational resources of benchmarking raises concerns about energy consumption and environmental sustainability, which should be taken into consideration when using the benchmark.

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

# A   Human Subjects Protocol and Informed Consent

Our research was thoroughly reviewed by the Research Review Committee (IRB equivalent) at our institute. The review process ensures that our proposed human subject protocol meets the highest standards with respect to Environment, Health and Safety, Ethics, Legal and Privacy guidelines. We also obtained informed consent from our participants who were provided a detailed document describing our study's protocol with clearly mentioned potential risks of the study. A study coordinator ensured that any questions participants had were answered prior to signing the consent form. Participants were informed that their participation was entirely voluntary and that they could decline to participate in all or any portion of the study at any time, for any reason. Participants were recruited through a vendor partner, using pre-approved recruitment and screening language and compensated in the amount of $50 per hour. We also mention in the consent agreement that the information collected may be used to author research papers.[5]

# B   Demographics of our data collectors

The table below describes the demographics of our study participants:

| Category | Group | Count |
|---|---|---|
| Gender | Male | 144 |
| | Female | 203 |
| | Prefer Not to Say | 1 |
| | Other | 2 |
| Ethnicity | South Asian | 41 |
| | Asian | 21 |
| | Caucasian | 132 |
| | Black / African American | 45 |
| | Mixed (Multi-racial) | 28 |
| | Hispanic or LatinX | 4 |
| | South East Asian | 18 |

Table 3: Combined Gender and Ethnicity Distribution

# C   Handling Privacy via Data Anonymization

We meticulously handled all personally identifiable information (PII) in our study. Faces and license plates were blurred to protect identities, and the dataset was designed with a scripted format. Participants were instructed to remove any visual or auditory details containing PII before data collection, and no other participants were present in the background. Each video underwent a thorough review by at least three annotators, and any content with potential PII was rejected and removed from the dataset. The study did not involve private conversations, and participants consented to a public data release. We highlighted the face blurring process in the paper and emphasized that the digital contexts were synthetic, ensuring no privacy violations.

# D   Dataset Statistics

Figure 7 visualizes distributions over several dimensions of our dataset, including:

- (Fig. 7a) **Goal types**. We can see that our dataset is skewed towards "Search" goal types (~2 thirds of all observation-goal pairs), given their generality and suitability.

- (Fig. 7b) **Script Description Word Cloud**. We took the short descriptions of each script (e.g., "'Troubleshoot a non-functioning leaf blower'") and plotted a word cloud showing the top 250 words, sized proportionally to the log of the count of the word. Themes emerge, such as memory, learning, health and fitness, meal preparation, daily chores, and recreation.

- (Fig. 7c) **Digital apps**. "Calculator" app is the most used in across observations (~10%), followed by "Search" (~4%) and "Messaging" (~3%).

- (Fig. 7d) **Density and diversity of recordings**.

---

[5]PDF of the human subjects protocol and consent documents can be found in our GitHub repository: https://github.com/facebookresearch/WAGIBench

(i) In terms of scripts, we can see from the histograms that the majority of scripts were recorded by 5-7 participants, while a small minority were recorded by as many as 15 participants. On the other hand, the distribution of number of videos per script is more uniform ranging from 1 to ~40 videos per script.

(ii) In terms of participants, we observe a concentration around 2-3 scripts per participant, while some participants recorded as many as 10 scripts. We also observe a concentration in participants who recorded <10 videos, while a small minority recorded as much as 40 videos.

- (Fig. 7e) **Modality volumes**. We compute histograms of observation volumes for all modalities. We observe histograms that exhibit bell-like shaped curves when plotted in logarithmic scale.

  V: video durations in seconds for vision,

  A: word counts for audio (we discard non-speech audio),

  D: digital states in kilobytes (KB)

  L: history data (vision + audio + digital) in kilobytes (KB)

We note that the video dataset we initially collected is 264 hours in length. When considering only videos that passed an initial quality review (considering the video, audio, and generated digital state quality), the size is reduced to 155 hours. Finally, applying the context-windowing (which eliminates context that is off-script) further reduces it to 29 hours.

| App Type | Description | Arguments |
|---|---|---|
| Search | Web or timeline search | source, query_item, query |
| Shop | Shopping | content |
| Store Memory | Grocery lists, todo list etc | content |
| Temporal Attention | Timers and reminders | action, time, content |
| Guided Activity | Workout videos, tutorials, etc | content, time |
| Communication | Messaging | action, target, content |
| Control Environment | IoT (lights, smart speaker, etc) | target, value |
| Translate | Audio or text translation | modality, src_language |
| Entertain | Playing music or videos | modality, content |

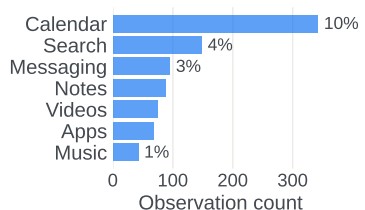

(a) (left) Table describing the types of digital goals in the dataset, and listing the arguments associated with each. (right) The number of videos of each type of goal in the dataset.

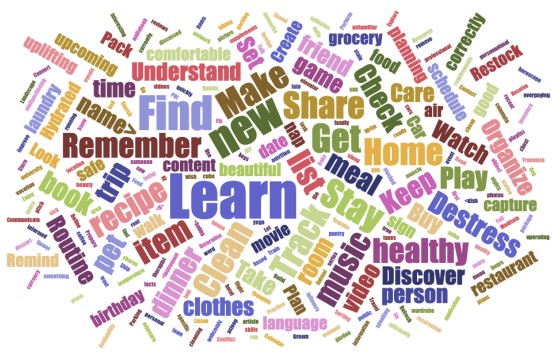

(b) Word cloud of script descriptions.

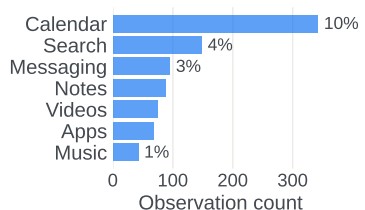

(c) Distribution of app annotations for generating digital context.

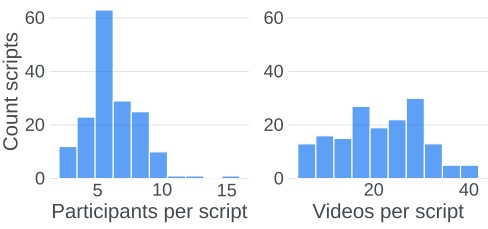

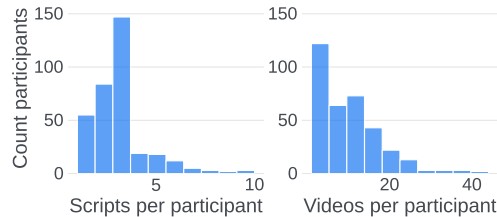

(d) Distributions of participants and videos per script (left), and of scripts and videos per participant (right).

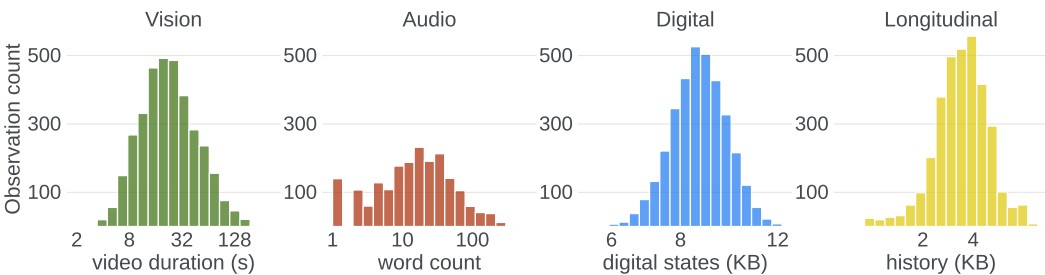

(e) Distributions over modality volumes (per-observation), in logarithmic scale (base-10 for audio and base-2 for the rest).

Figure 7: Overall dataset statistics



Figure 8: Goal inference examples with vision-only contexts. Best viewed when zoomed in.

# E  Qualitative Results

## E.1  Generative Examples

In this section we show examples of goal inference from various modalities and analyze where different input modalities help or fail.

### E.1.1  Vision-only Context Examples

In Fig. 8, we present several examples showing the model's (Qwen2.5-VL-72B) prediction using only vision-contexts (i.e. video frames). All three examples are drawn from $S_V$ subset which means supposedly these goals can be inferred with vision contexts only. With no surprise, all three predicted goals are considered very relevant by the LLM Judge, despite for the center and right example, they differ from the reference goals.

### E.1.2  Audio Context Examples

Fig. 9 shows three examples from the subset $S_{VA}$ where Qwen2.5-VL-72B's predictions with V inputs are irrelevant, while predictions with VA inputs are very relevant. These examples, which feature varied locations and lighting settings, illustrate the importance of the audio modality in cases where vision alone can be very misleading due to the lack of relevant contextualized information present in audio transcriptions. The audio transcriptions shown in the figure may be subject to spelling errors due to limitations in Automatic Speech Recognition (ASR) system used for transcription.

### E.1.3  Digital Context Examples

Fig. 10 shows three examples from the subset $S_{VD}$ from Qwen2.5-VL-72B using multiple input modalities V, VD and VD$^*$. In Fig. 10a, we see the typical case where visual and digital modalities are well-aligned and additional information in VD and VD$^*$ modalities helps predict the goal accurately over the V modality. Fig. 10b shows a failure case where both vision and digital states have many distractors and the model is not able to accurately predict the goal, except by using the high-signal VD$^*$ modality. Finally, Fig. 10c shows a case where the visual modality dominates the model's prediction and even with VD$^*$, it is hard to predict the correct digital goal as the model focuses on the incorrect visual distractor (Christmas tree in the background).

### E.1.4  Longitudinal Context Examples

Fig. 11 shows three examples from the subset $S_{VL}$ from Qwen2.5-VL-72B using multiple input modalities V, VL and VL$^*$. In Fig. 11a, we see how longitudinal history assists VL and VL$^*$ modalities to personalize their user goal prediction relative to the V modality. In Fig. 11b, potentially due to the high amount of distracting user history in the VL modality, we see how the model is unable to accurately predict a personalized goal. Only the high-signal VL$^*$ modality that is void of distracting history observations uses both video-specific visual cues (about to run out of bread) and longitudinal history (of the user adding pasta to the grocery list) to recommend adding bread to the grocery list. Fig. 11c shows a case where the visual cues override longitudinal history and shows the model's prediction being relevant to organizing a cluttered space. But, both VL and VL$^*$ input modalities fail to identify that the user likes to listen to music while de-cluttering their space.

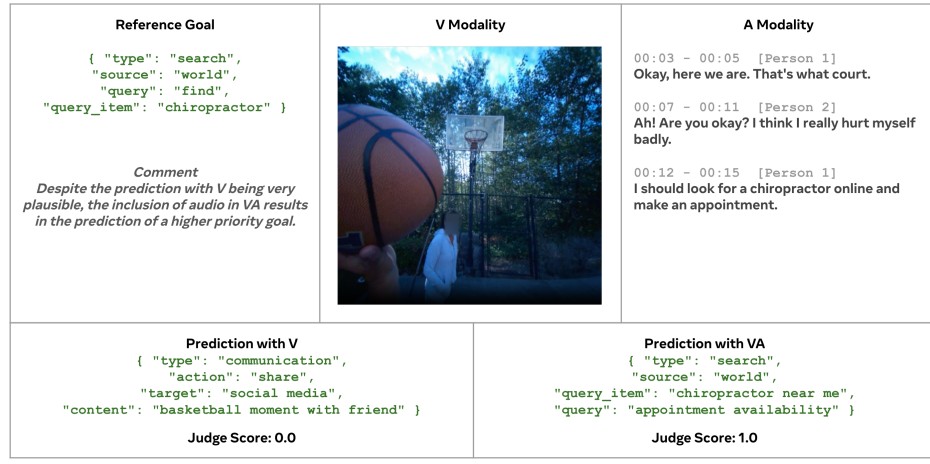

(a) Outdoors example.

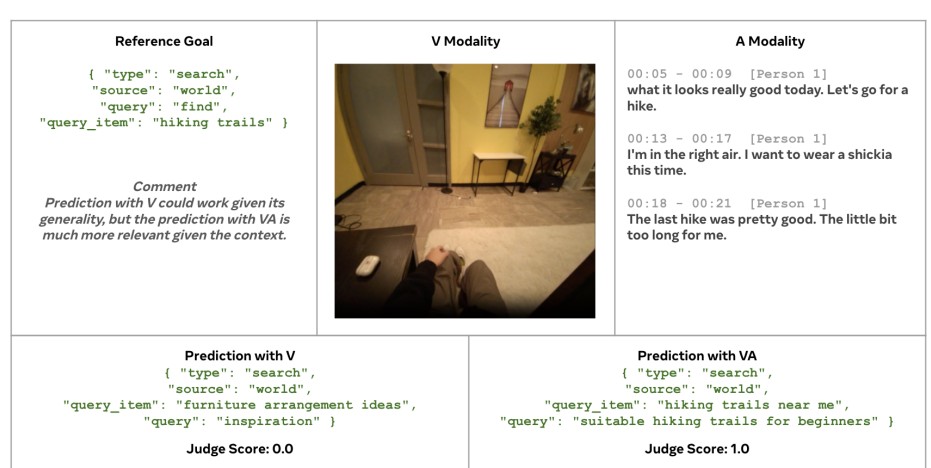

(b) Indoors example with good lighting conditions.

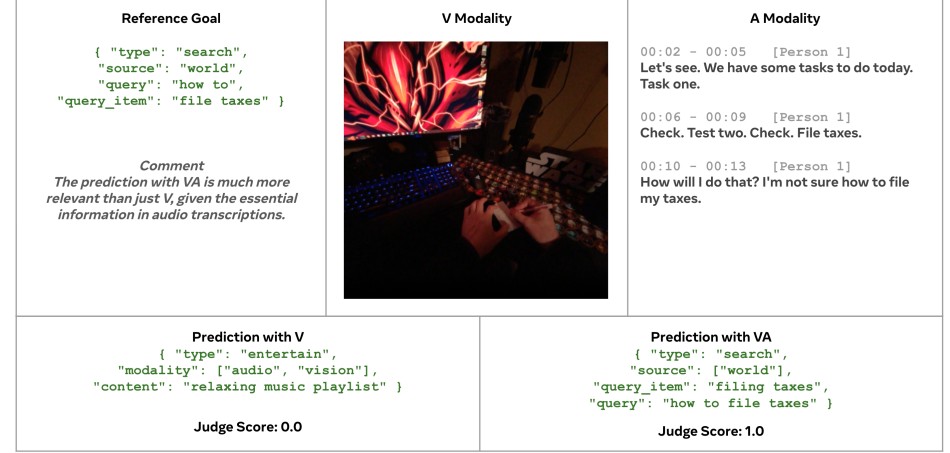

(c) Indoors example with darker lighting conditions.

Figure 9: Qualitative examples from the audio modality.

**Reference Goal**

```
{'type': 'search', 'source':
'world', 'query': 'nearby movie
times for', 'query_item': 'inside
out 2'}
```

**Cues**
{'Calendar': 'several upcoming calendar events for going to see movies.', 'Videos': 'The user opens YouTube and plays inside out 2 trailer.'}

*Comment*
*V fails to identify the movie, but both VD and VD\* are able to predict meaningful goals for the correct movie based on the digital state*

**V Modality**

**D\* Modality**

```
'current_datetime':
'2025-04-21 17:57'

'Calendar':
[...,
    {'title': 'Cinema Evening with
Caleb Hall', 'start_datetime':
'2025-04-23 18:00',
'end_datetime': '2025-04-23
21:00', 'location': 'Landmark
Theatres, 911 Pine St, Seattle',
...}
]

'Videos':
[...,
    {'title': 'Inside Out 2
Official Trailer',
'duration_secs': 147, 'category':
'Movies', 'watch_timestamp':
'2025-04-20 20:45:00'}
]
```

**Prediction with V**
```
{"type": "search", "source":
["world"], "query_item":
"Everything Movie", "query":
"release date and cast"}
```

**Judge Score: 0.0**

**Prediction with VD**
```
{"type": "search", "source":
["world"], "query_item": "Inside
Out 2", "query": "showtimes and
ticket availability"}
```

**Judge Score: 1.0**

**Prediction with VD\***
```
{"type": "search", "source":
["world"], "query_item": "Inside
Out 2", "query": "release date and
cast"}
```

**Judge Score: 1.0**

(a) Both VD and VD* correctly predict the user's goal

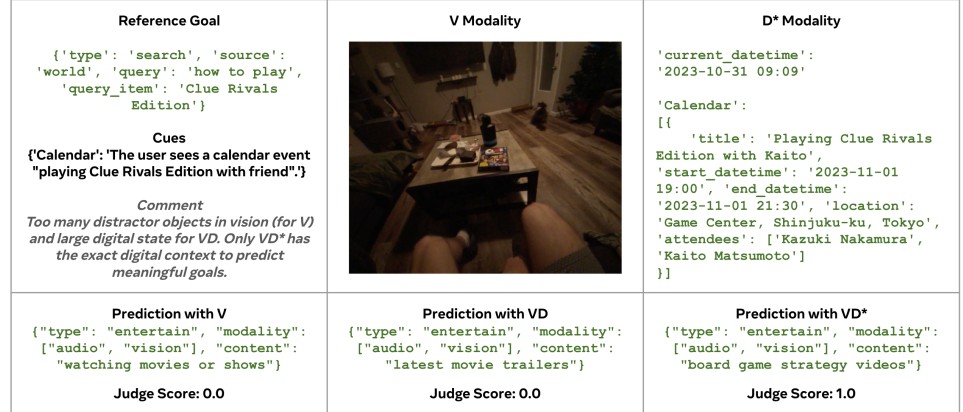

**Reference Goal**

```
{'type': 'search', 'source':
'world', 'query': 'how to play',
'query_item': 'Clue Rivals
Edition'}
```

**Cues**
{'Calendar': 'The user sees a calendar event "playing Clue Rivals Edition with friend".'}

*Comment*
*Too many distractor objects in vision (for V) and large digital state for VD. Only VD\* has the exact digital context to predict meaningful goals.*

**V Modality**

**D\* Modality**

```
'current_datetime':
'2023-10-31 09:09'

'Calendar':
[{
    'title': 'Playing Clue Rivals
Edition with Kaito',
'start_datetime': '2023-11-01
19:00', 'end_datetime':
'2023-11-01 21:30', 'location':
'Game Center, Shinjuku-ku, Tokyo',
'attendees': ['Kazuki Nakamura',
'Kaito Matsumoto']
}]
```

**Prediction with V**
```
{"type": "entertain", "modality":
["audio", "vision"], "content":
"watching movies or shows"}
```

**Judge Score: 0.0**

**Prediction with VD**
```
{"type": "entertain", "modality":
["audio", "vision"], "content":
"latest movie trailers"}
```

**Judge Score: 0.0**

**Prediction with VD\***
```
{"type": "entertain", "modality":
["audio", "vision"], "content":
"board game strategy videos"}
```

**Judge Score: 1.0**

(b) Only VD* correctly predicts the user's goal

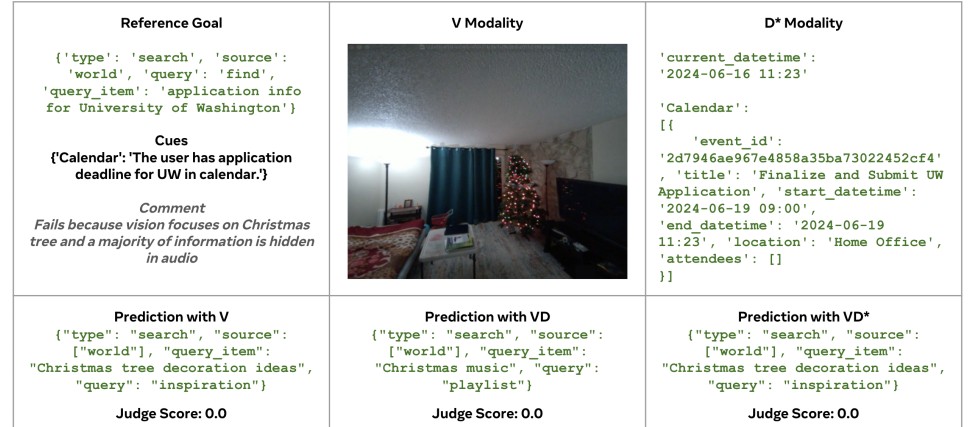

**Reference Goal**

```
{'type': 'search', 'source':
'world', 'query': 'find',
'query_item': 'application info
for University of Washington'}
```

**Cues**
{'Calendar': 'The user has application deadline for UW in calendar.'}

*Comment*
*Fails because vision focuses on Christmas tree and a majority of information is hidden in audio*

**V Modality**

**D\* Modality**

```
'current_datetime':
'2024-06-16 11:23'

'Calendar':
[{
    'event_id':
'2d7946ae967e4858a35ba73022452cf4'
, 'title': 'Finalize and Submit UW
Application', 'start_datetime':
'2024-06-19 09:00', 'end_datetime':
'2024-06-19 11:23', 'location':
'Home Office', 'attendees': []
}]
```

**Prediction with V**
```
{"type": "search", "source":
["world"], "query_item":
"Christmas tree decoration ideas",
"query": "inspiration"}
```

**Judge Score: 0.0**

**Prediction with VD**
```
{"type": "search", "source":
["world"], "query_item":
"Christmas tree decoration", "query":
"playlist"}
```

**Judge Score: 0.0**

**Prediction with VD\***
```
{"type": "search", "source":
["world"], "query_item":
"Christmas tree decoration ideas",
"query": "inspiration"}
```

**Judge Score: 0.0**

(c) Neither VD nor VD* correctly predicts the user's goal

Figure 10: Qualitative examples from the digital modality

| Reference Goal | V Modality | L* Modality |
|---|---|---|
| {'type': 'guided_activity', 'content': 'white noise', 'time': '0:01:48'} | 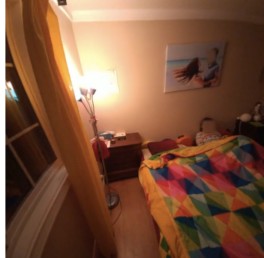 | Context: The video captures a serene bedtime routine. The person adjusts the bedding and prepares for sleep. The lamp casts a warm glow, highlighting the room's cozy ambiance. The person wears a watch, indicating their presence in the scene. |
| *Comment*
*VL and VL\* combine user history and visual observations to accurately predict the goal.* | | Action: {'type':'guided_activity', 'content': 'white noise', 'time': '0:02:06'} |
| **Prediction with V**
{ "type": "search", "source": ["world"], "query_item": "lamp design", "query": "inspiration for modern lamp designs" }

**Judge Score: 0.0** | **Prediction with VL**
{ "type": "guided_activity", "content": "relaxation exercise", "time": "15 minutes" }

**Judge Score: 1.0** | **Prediction with VL\***
{ "type": "entertain", "modality": ["audio"], "content": "relaxing music"}

**Judge Score: 1.0** |

(a) Both VL and VL* correctly predict the user's goal

| Reference Goal | V Modality | L* Modality |
|---|---|---|
| {'type': 'store_memory', 'content': 'add bread to grocery list'} | 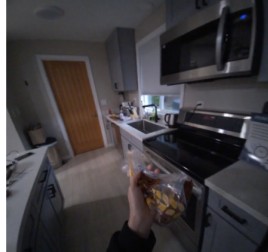 | Context: The setting includes typical kitchen items, highlighting a cooking environment. The video shows a kitchen scene with a stove, pots, and a cutting board. A hand interacts with organic food items on the counter. |
| *Comment*
*V and VL make generic predictions about retrieving food from the refrigerator. VL\* enhances this by incorporating past history and current low bread stock to add items to the shopping list.* | | Action: {'type': 'store_memory', 'content': 'add pasta to grocery list'} |
| **Prediction with V**
{ "type": "search", "source": ["world"], "query_item": "banana", "query": "recipes"}

**Judge Score: 0.0** | **Prediction with VL**
{ "type": "store_memory", "content": "I retrieved a packaged item from the refrigerator and placed it on the counter" }

**Judge Score: 0.0** | **Prediction with VL\***
{"type": "store_memory", "content": "add bread and milk to grocery list"}

**Judge Score: 0.5** |

(b) Only VL* correctly predicts the user's goal

| Reference Goal | V Modality | L* Modality |
|---|---|---|
| {'type': 'entertain', 'modality': 'audio', 'content': 'song playlist'} | 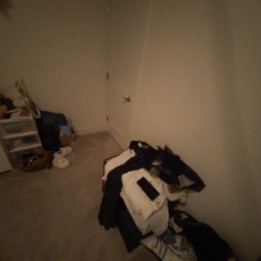 | Context: In a dimly lit garage, a person moves around and inspects the area filled with parked cars and scattered items. The setting suggests a private residential space used for both parking and storage, highlighting a mundane, everyday scenario. |
| *Comment*
*V makes irrelevant prediction. VL and VL\* identify a cluttered space making relevant predictions. But they fail to personalize from history that the user likes to listen to music while decluttering.* | | Action: {'type': 'entertain', 'modality': 'audio', 'content': 'song playlist'} |
| **Prediction with V**
{ "type": "search", "source": ["world"], "query_item": "Automatic Storage Water Heater", "query": "installation instructions" }

**Judge Score: 0.0** | **Prediction with VL**
{"type": "search", "source": "world", "query_item": "storage solutions", "query": "best ways to organize a cluttered home"}

**Judge Score: 0.0** | **Prediction with VL\***
{ "type": "search", "source": ["world"], "query_item": "home organization tips", "query": "ways to declutter and organize a home"}

**Judge Score: 0.0** |

(c) Neither VL nor VL* correctly predicts the user's goal

Figure 11: Qualitative examples from the longitudinal modality

## E.2 Comparing Human to Model Performance via MCQ Task Examples

To qualitatively compare human and model performance, we present a few sets of MCQ examples from the human predictability study in Figures 12, 13, 14, 15. Each row represents one MCQ problem. A relevant frame from the video is shown on the left. On the right is a block of text containing (1) a *description* of what happens in the video, (2) any transcribed *speech* from the audio, (3) the average human and model (across all tested models) accuracies, (4) the *reference goal*, (5) (in green) MCQ options that humans selected, (6) (in purple) MCQ options that models selected, and (7) (in gray) the full set of MCQ options.

Figure 12 contains "easy" examples which both humans and models are able to predict with high accuracy. Figure 13 contains examples where humans appear to have strong intuitions about which goals may be relevant, but models fail. Figure 14 contains examples where there are *multiple relevant goals* in the option-set (examples where the strong reference bias of MCQ introduces noise into the evaluation). Figure 15 contains examples that require *fine-grained* visual recognition (e.g., reading text or identifying small objects like house plants) in order to solve, where models often struggle.

### MCQ Examples Where Models Succeed

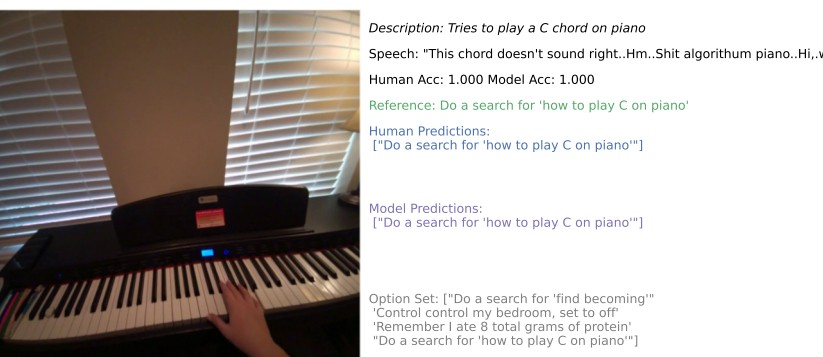

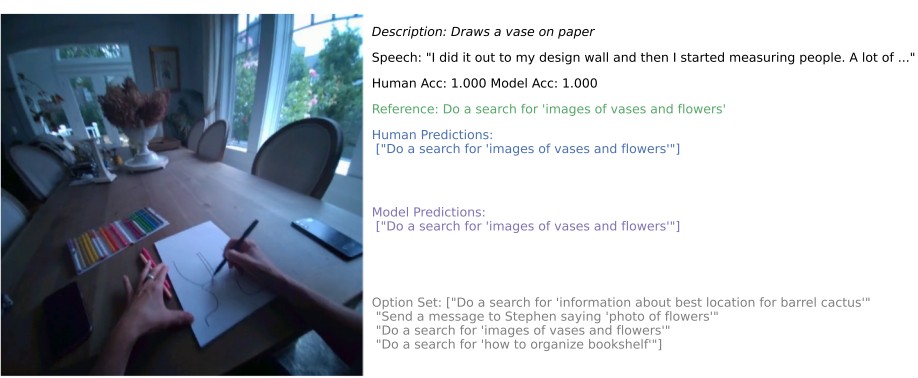

Figure 12: See Section E.2 in the text for a full description of the figure.

# MCQ Examples where Models Struggle to Intuit Goals

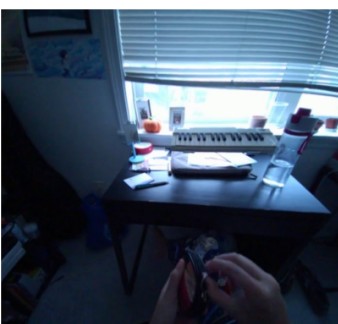

*Description: Person looks at laundry and checks coin purse*
Speech: "there."
Human Acc: 1.000 Model Acc: 0.500
Reference: Do a search for 'location of laundromat'
Human Predictions:
 ["Do a search for 'location of laundromat'"]

Model Predictions:
 ["Do a search for 'how to clean bagless vacuum cleaner'"
 "Do a search for 'location of laundromat'"]

Option Set: ["Do a search for 'how to clean bagless vacuum cleaner'"
 'Remember remember my car is located here'
 "Do a search for 'information about indigo urgent care'"
 "Do a search for 'location of laundromat'"]

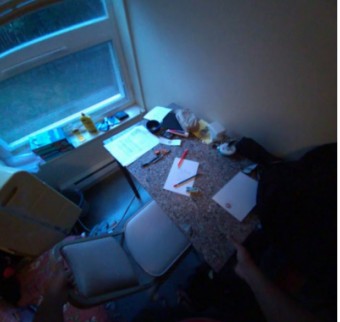

*Description: Person looks around their cramped space and tries to move a chair*
Speech: "man..Stu long.has set to.congress.the good"
Human Acc: 1.000 Model Acc: 0.500
Reference: Do a search for 'how to setup small living space'
Human Predictions:
 ["Do a search for 'how to setup small living space'"]

Model Predictions:
 ["Do a search for 'how to setup small living space'"
 "Do a search for 'how to install frame'" 'Control dining room, set to on']

Option Set: ["Do a search for 'how to setup small living space'"
 "Do a search for 'how to install frame'" 'Control dining room, set to on'
 "Do a search for 'strategies for separating clothes for laundry'"]

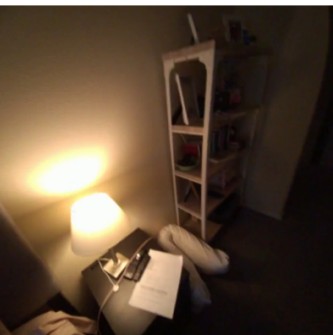

*Description: Person looks at the lamp*
Speech: ""
Human Acc: 1.000 Model Acc: 0.750
Reference: Control lamp, set to off
Human Predictions:
 ['Control lamp, set to off']

Model Predictions:
 ['Control lamp, set to off'
 "Send a message to Brianna Clark saying 'Ed Sheeran - Shape of You'"]

Option Set: ["Do a search for 'recipes with marinara sauce, spaghetti, butter'"
 "Send a message to Brianna Clark saying 'Ed Sheeran - Shape of You'"
 'Launch a guided activity to do run near current location with a duration of 0:16:40'
 'Control lamp, set to off']

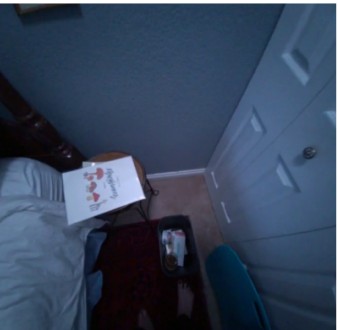

*Description: Person looks at recycling container*
Speech: ""
Human Acc: 0.667 Model Acc: 0.500
Reference: Do a search for 'information about recycling schedule'
Human Predictions:
 ["Do a search for 'information about recycling schedule'"
 'Control control my bedroom, set to off']

Model Predictions:
 ['Remember 3 cheese nachos'
 "Do a search for 'information about recycling schedule'"
 "Do a search for 'information about Justin Jefferson'"]

Option Set: ["Do a search for 'information about Justin Jefferson'"
 'Control control my bedroom, set to off' 'Remember 3 cheese nachos'
 "Do a search for 'information about recycling schedule'"]

Figure 13: See Section E.2 in the text for a full description of the figure.

# MCQ Examples with Multiple Relevant Goals

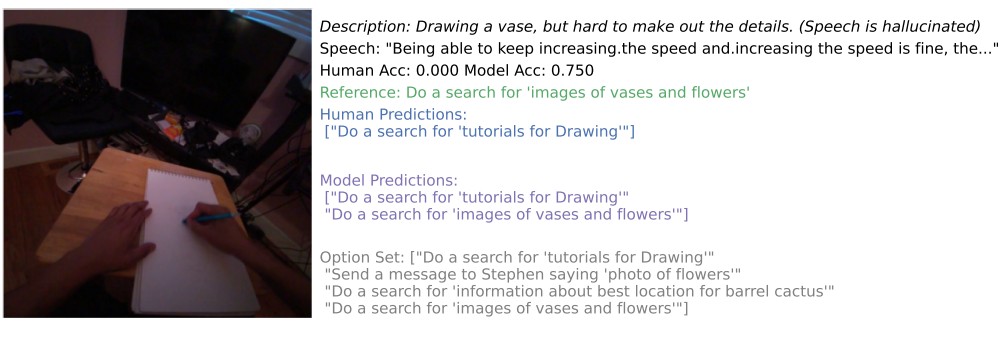

*Description: Drawing a vase, but hard to make out the details. (Speech is hallucinated)*
Speech: "Being able to keep increasing.the speed and.increasing the speed is fine, the..."
Human Acc: 0.000 Model Acc: 0.750
Reference: Do a search for 'images of vases and flowers'
Human Predictions:
 ["Do a search for 'tutorials for Drawing'"]

Model Predictions:
 ["Do a search for 'tutorials for Drawing'"
 "Do a search for 'images of vases and flowers'"]

Option Set: ["Do a search for 'tutorials for Drawing'"
 "Send a message to Stephen saying 'photo of flowers'"
 "Do a search for 'information about best location for barrel cactus'"
 "Do a search for 'images of vases and flowers'"]

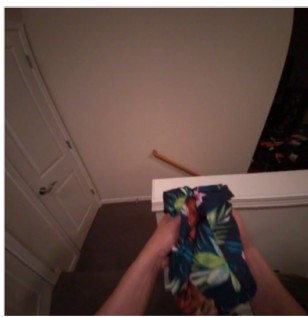

*Description: Person picks up a shirt and examines it*
Speech: ""
Human Acc: 0.667 Model Acc: 0.000
Reference: Do a search for 'how to hand wash delicate clothing'
Human Predictions:
 ['Remember Add fold laundry to todo list (complete by 8:00 PM)'
 "Do a search for 'how to hand wash delicate clothing'"]

Model Predictions:
 ['Remember Add fold laundry to todo list (complete by 8:00 PM)']

Option Set: ["Do a search for 'how to hand wash delicate clothing'"
 "Do a search for 'how to display utensils'" 'Shop for body wash'
 'Remember Add fold laundry to todo list (complete by 8:00 PM)']

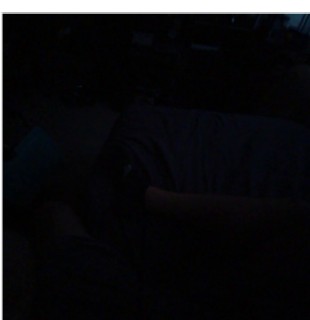

*Description: Person wakes up and complains of being tired*
Speech: "Oh boy,.I still feel.pretty tired."
Human Acc: 0.667 Model Acc: 0.000
Reference: Launch a guided activity to do Analyze sleep patterns with a duration of 10 minutes
Human Predictions:
 ['Launch a guided activity to do Analyze sleep patterns with a duration of 10 minutes'
 "Set a timer/reminder for 30 with the note 'Wake up timer'"]

Model Predictions:
 ["Set a timer/reminder for 30 with the note 'Wake up timer'"]

Option Set: ['Launch a guided activity to do guided meditation with a duration of 0:00:00'
 "Set a timer/reminder for 30 with the note 'Wake up timer'"
 'Launch a guided activity to do Analyze sleep patterns with a duration of 10 minutes'
 'Launch a guided activity to do run near current location with a duration of 0:16:40']

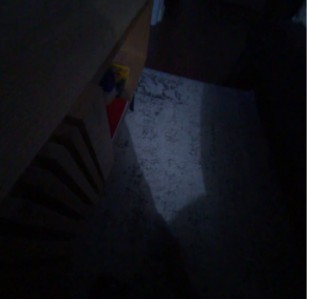

*Description: Person rubs the carpet indicating a stain*
Speech: ""
Human Acc: 1.000 Model Acc: 0.750
Reference: Do a search for 'how to remove stain'
Human Predictions:
 ["Do a search for 'how to remove stain'"]

Model Predictions:
 ["Do a search for 'how to troubleshoot broken vacuum cleaner'"
 "Do a search for 'how to remove stain'"]

Option Set: ["Do a search for 'how to troubleshoot broken vacuum cleaner'"
 "Do a search for 'how to cut melon'"
 "Do a search for 'how to display plushes'"
 "Do a search for 'how to remove stain'"]

Figure 14: See Section E.2 in the text for a full description of the figure.

# MCQ Examples where Models Struggle with Perception

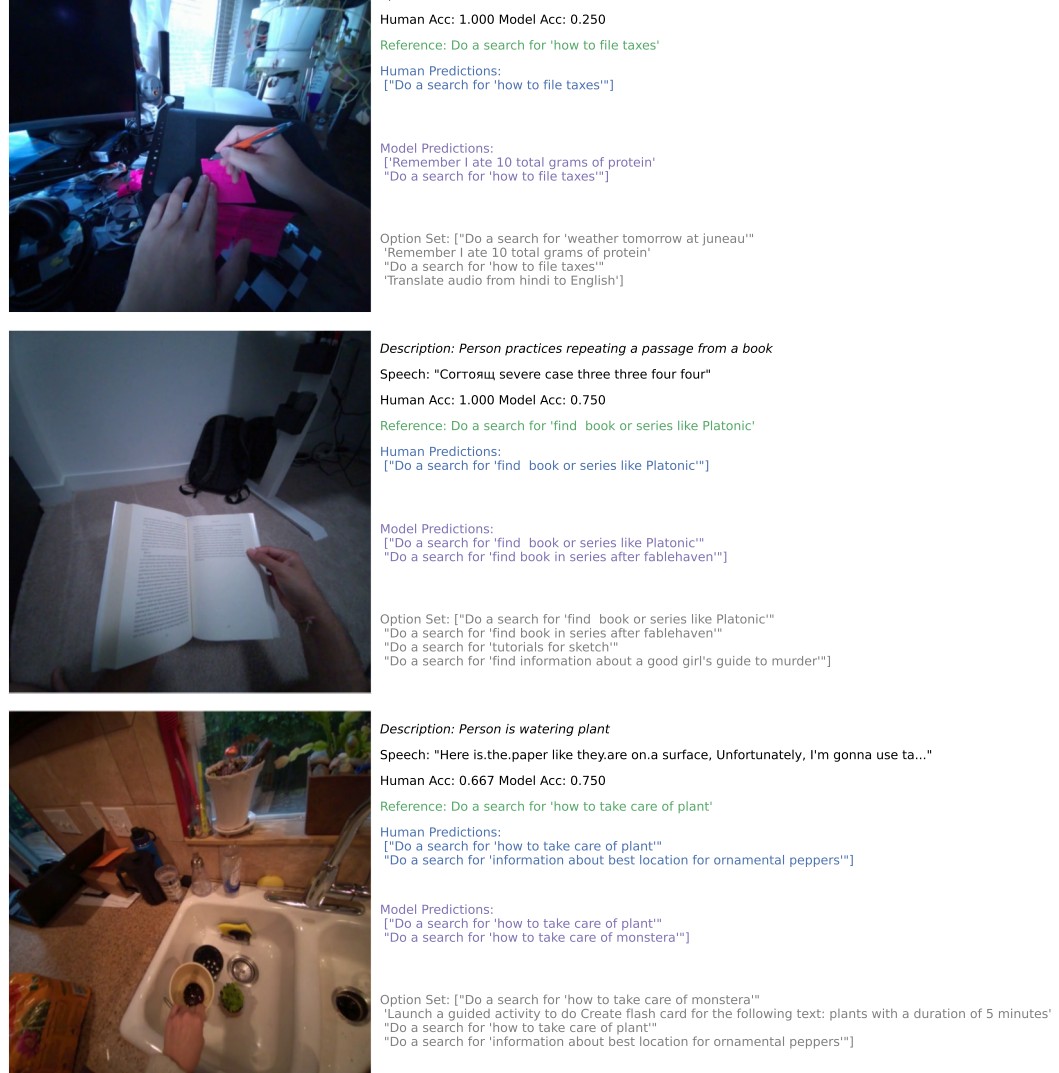

*Description: Person crosses off items on the list until reaching 'file taxes'*

Speech: ""

Human Acc: 1.000 Model Acc: 0.250

Reference: Do a search for 'how to file taxes'

Human Predictions:
["Do a search for 'how to file taxes'"]

Model Predictions:
['Remember I ate 10 total grams of protein'
"Do a search for 'how to file taxes'"]

Option Set: ["Do a search for 'weather tomorrow at juneau'"
'Remember I ate 10 total grams of protein'
"Do a search for 'how to file taxes'"
'Translate audio from hindi to English']

*Description: Person practices repeating a passage from a book*

Speech: "Согтоящ severe case three three four four"

Human Acc: 1.000 Model Acc: 0.750

Reference: Do a search for 'find  book or series like Platonic'

Human Predictions:
["Do a search for 'find  book or series like Platonic'"]

Model Predictions:
["Do a search for 'find  book or series like Platonic'"
"Do a search for 'find book in series after fablehaven'"]

Option Set: ["Do a search for 'find  book or series like Platonic'"
"Do a search for 'find book in series after fablehaven'"
"Do a search for 'tutorials for sketch'"
"Do a search for 'find information about a good girl's guide to murder'"]

*Description: Person is watering plant*

Speech: "Here is.the.paper like they.are on.a surface, Unfortunately, I'm gonna use ta..."

Human Acc: 0.667 Model Acc: 0.750

Reference: Do a search for 'how to take care of plant'

Human Predictions:
["Do a search for 'how to take care of plant'"
"Do a search for 'information about best location for ornamental peppers'"]

Model Predictions:
["Do a search for 'how to take care of plant'"
"Do a search for 'how to take care of monstera'"]

Option Set: ["Do a search for 'how to take care of monstera'"
'Launch a guided activity to do Create flash card for the following text: plants with a duration of 5 minutes'
"Do a search for 'how to take care of plant'"
"Do a search for 'information about best location for ornamental peppers'"]

Figure 15: See Section E.2 in the text for a full description of the figure.

# F Human experiment trial structure

The figures below show the trial design corresponding to a single trial of the MCQ human study (Fig. 16a) and the meta-evaluation human study (Fig. 16b).

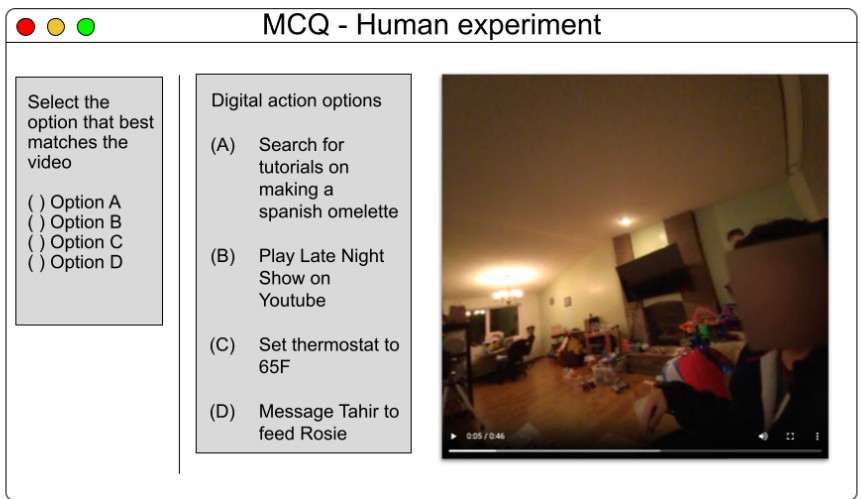

(a) Example structure of a single trial in the human MCQ experiment

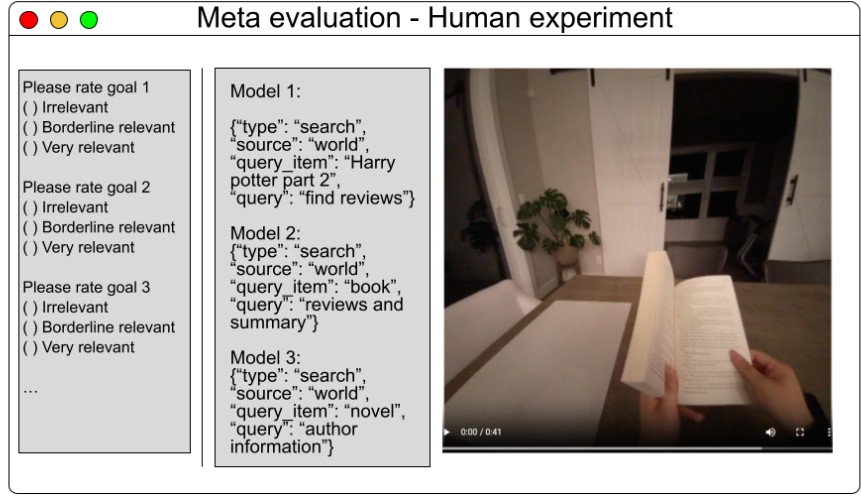

(b) Example structure of a single trial in the human meta evaluation experiment

# G  Modality-Specific Details

## G.1  Digital Context Generation

As mentioned briefly in main text, we designed a pipeline for generating rich digital contexts representing the internal app states of seven widely-used apps: *Calendar*, *Messaging*, *Notes*, *Search*, *Videos*, *Maps*, and *Music*.

For this, we associated the relevant contextual cues of *digital* modality for each scenario with an app from the above seven apps. This resulted in 825 observation-goal pairs across 43 scenario scripts having at least one digital cue annotated with one of the seven apps. We show the resulting annotated app distribution in Figure 7c. For the remaining 2,652 observation-goal pairs without relevant digital cues, we generate digital context without any cue-conditioning.

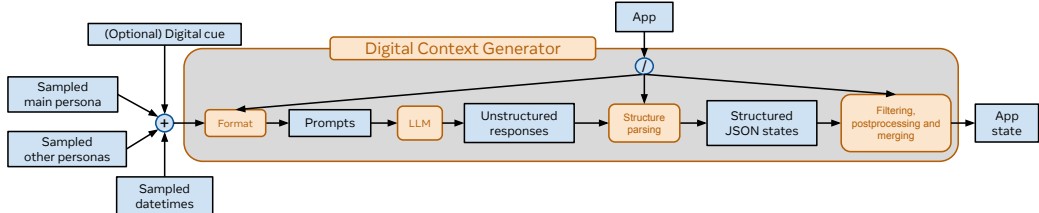

Figure 17: Digital Context Generator pipeline: A sampled main user persona, other personas, current datetime, and optionally a digital cue are processed via the DCG to generate structured app states.

The core of this system is a module we term the **Digital Context Generator (DCG)**, which synthesizes app-specific internal states from high-level persona and digital cues using a Large Language Model (LLM). Towards this we build a collection of 50 human personas containing synthesized names, ages, genders, nationalities, occupations and other useful fields. As illustrated in Figure 17, the DCG takes as input a sampled main user persona, 5 other related personas (e.g., friends, co-workers), a sampled current datetime and optionally a contextual digital cue. The process to generate the app states follows these stages:

1. **Formatting:** Inputs are filled into prompt templates with instructions to generate digital states for a given app. The prompt templates are crafted to reflect realistic app usage patterns. The input personas and the sampled datetime help to generate app states uniquely tailored to the scenario under consideration. If an optional digital context cue is provided, it is treated by a separate prompt for the relevant app since encoding a cue can require following specialized instructions for each app.

2. **LLM Inference:** The prompts are fed to an LLM (we used *Llama3.3-70b-instruct* model for this purpose), which outputs unstructured text describing plausible digital activity and app interactions for the user.

3. **Structure Parsing:** The unstructured LLM output is then parsed into structured JSON representations specific to each app's internal data structures.

4. **Filtering, Postprocessing and Merging:** Structured states are filtered to reject or correct invalid values for fields of the internal data structures. If a digital cue was provided to an app, that app separately generates sub-states derived from the digital contextual cue. These are merged with the app sub-states generated without the cue into a unified data structure. This final output is a coherent snapshot of the user's internal app state at the sampled time.

This approach enables the generation of diverse and human-like app states across a variety of temporal, personal, and situational contexts. These app states are semantically coherent, temporally relevant, and contextually grounded in the persona's attributes and environment. We produce a realistic digital state for all apps for all the 3,477 videos in the benchmark even if they do not have an associated digital cue. This results in an average of about 8.9k characters of digital state input for each video for a total of about 31 million digital state characters across the full dataset. Note that this implies that digital context contains a lot of distractor information, which models must ignore to correctly infer the user's goals.

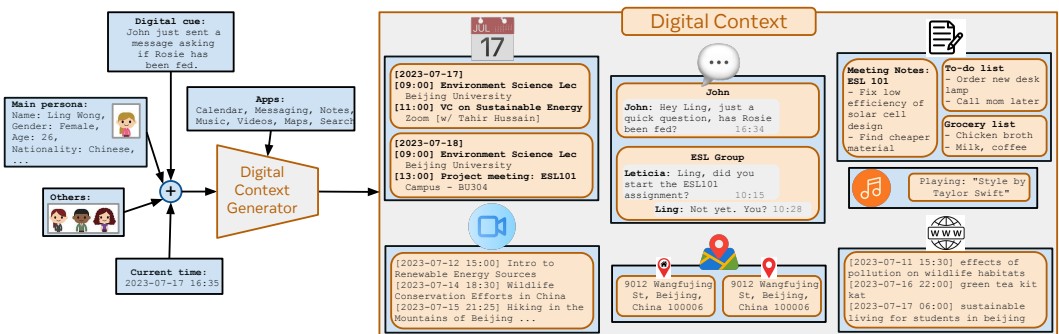

Figure 18: Example digital app states generated by the DCG: The main persona is "Ling", a Beijing University student, with interests in environmental science and sustainable energy. The digital state reflects a snapshot of Ling's app usage on a weekday evening. Note that the digital cue *John just sent a message asking if Rosie has been fed* is reflected in Ling's Messaging app state.

An example of a fully generated digital state is shown in Figure 18 and demonstrates the DCG's capacity to produce complex digital traces that mirror real-world app states.

## G.2  Vision and Audio Processing Details

For all models except Llama models, we uniformly sample 32 frames from the context video as the input to the model. For Llama-3.2, we sample 1 frame since it's only trained for single-image understanding. For video frame resolutions, we follow the default as much as possible which leads to input resolution 448x448 for GPT-4.1 and InternVL models, 700x700 for Qwen2.5 models and 1120x1120 for Llama-3.2, respectively.

To generate audio contexts, we use Whisper-base model [16] to transcribe all videos with a beam size of 5, and a temperature schedule of 0, 0.2, 0.4, 0.6, 0.8, and 1.0. We separately perform speaker diarisation and voice activity detection using internal models and add them to the transcription. Meanwhile, we pre-process the dataset to automatically generate captions for all videos using a set of VLMs, and perform filtering by checking the consensus of different VLM models. These video captions are then used as the longitudinal history contexts for evaluation.

## G.3  Longitudinal History Details

In this section we elaborate on the procedure to generate history context cues for the longitudinal scenarios as defined in the main paper.

**History Bank Generation.** For each video corresponding to a longitudinal scenario, we sampled a history bank of 5 support videos from recordings of the same participant and environment. One of these support videos shares the same script as the test video; this corresponds to positive longitudinal history. The other videos which don't share the test video's script are distractors that are used to reflect an ecologically valid longitudinal history setup.

**Textual representation of audiovisual history context** We represent longitudinal history in the form of detailed video captions and audio transcriptions of support videos. video captions as longitudinal history contexts. For each 10-second clip of a support video, first we use Qwen2.5-VL-72B and InternVL2.5-78B-MPO to separately generate generate video summarization. We then leverage DeepSeek-R1-Distill-Llama-70B to process the two generations and merge them into one video summary. The LLM merger is instructed to remove any information that exists only in the generation from one models, and only keep the description that are shared across both VLM models. Once we obtain summaries for all 10s chunks of the support video, we concatenate those that fall within the context window annotation as a representation of the visual support context from this video. We note it is possible that this captioning process introduces noise into the representation of longitudinal history videos. However, we note that our use of the abovementioned LLM merger reduces the likelihood of such false information occurring in the captions. In addition, we also include the audio transcription of the video in the history.

**Longitudinal evaluation of VLMs** As shown below in Section I.1, each support video's history context and structured goal annotation are packaged into a JSON dictionary and shuffled along with other support videos for a given test video during evaluation.

# H  Meta-Evaluation Implementation Details

In this section, we provide the implementation details for our Meta-Evaluation experiments designed to study the alignment between LLM Judge models and human raters.

For each generated goal, we assign three raters to assess its quality. Among them, results from the first two raters are used to obtain the "ground-truth" pairwise relative ranking order, while the third one is held out and later used to compute human-human alignment (we'll elaborate this next). We loop over all possible pairs of predicted goals given a single video and filter out pairs of goals for which two human raters do not agree on their relative order (e.g., $<$ and $>$ are considered disagreeing each other, whereas $=$ and $<$ are not). We choose to filter goal pairs instead of individual goals as it's much easier for humans to agree on the relative ordering of goals compared to giving consistent absolute scores. With this filtering, we end up removing 2.3% of model pairs where their rankings are inconsistent between raters. The filtered set is then used as ground-truth to compute ranking accuracy for judge models.

Specifically, for each video, we loop over all pairs in the filtered ground-truth set, and compute the ranking accuracy for the LLM Judge against the ground-truth relative ranking. For scenarios when LLM judge gives tied ranking to two VLM outputs, we consider the LLM judge is aligned with ground-truth if at least one human raters give the tied ranking. When using sentenceBERT as the judge, we treat two VLM outputs as tied if their sentenceBERT score difference is within 0.1. Finally, to measure the human-human alignment, we take the rating from the third human rater and compute its alignment to ground-truths in the same way as we did for LLM Judge models.

# I  Modeling Details

## I.1  Example Prompts for Goal Inference and LLM Judge

First, we present the prompt for VLMs to answer multi-choice questions.

---
**MCQ Prediction Prompt**

```
You are an intelligent assistant.  You will be given a video and four options, each
containing a digital action formatted as a dictionary.  Your task is to respond with
which option (A or B or C or D) is the most likely digital action that follows a
human's context window of actions present in the video.

{mcq_options}

NOTE:
- Your response should contain only the option letter A, B, C, or D. Only respond with
one letter.  Do not repeat the option.
- Wrap your final answer with <answer> and </answer> tag.  For instance, an example full
output should look like this:  <answer>A</answer>
```
---

In addition to this base MCQ prompt, we optionally augment it with various modality contexts (i.e., audio, digital, longitudinal). Below we show examples for each of these context modalities.

---
**Audio Contexts Example**

```
To facilitate the task, the transcription for the video is provided as follows.
One of the speakers may be wearing smart glasses. Feel free to ignore the transcription
if it is not relevant. The transcription is:

Speaker 0: Let's see. We have some tasks to do today. Task one.
Speaker 0: Check. Test two. Check. File taxes.
Speaker 0: How will I do that? I'm not sure how to file my taxes.
```
---

---
**Digital Contexts Example**

```
The current datetime is: ['2023-10-07 12:05']

Current state of the user's Calendar app:
[...,
{'event_id': ['30a546df10154a99a302ca37d12d0b97'],
 'title': ['Playing Cards Against Humanity with friend'],
 'start_datetime': ['2023-10-07 20:00'],
 'end_datetime': ['2023-10-07 23:00'],
 'location': ["Javier's home, 3456, Calle de la Luna, Madrid, Spain 28004"],
```
---

```
  ...
]

Current state of the user's Messaging app:
[{'participants': [('Javier Garcia',), ('Alexander Brooks',)],
  'messages': [
    {'sender': ['Javier Garcia'],
     'message_id': ['6b228dbb2c54412f805ae28f7e76ac3b'],
     'timestamp': ['2023-09-20 10:00'],
     'content': ["Alexander, hope you're doing well. I've been meaning to ask, have you
     had a chance to read 'The Shadow of the Wind' by Carlos Ruiz Zafón? I think you'd
     appreciate the themes of literature and mystery."]},
    {'sender': ['Alexander Brooks'],
     'message_id': ['27fc1cdbf4ac47b5a0431103cf779ea8'],
     'timestamp': ['2023-09-22 15:30'],
     'content': ['Javier, thanks for the recommendation! I actually just finished reading
     it and loved it. The way Zafón weaves the story around the Cemetery of Forgotten
     Books is captivating.']}
  ],
  'title': ['Alexander Brooks'],
  'conversation_id': ['f112cec95ee0448ea5301c71a7077124']
},
...
]

Current state of the user's Search app:
[{'query': ['Spanish grammar exercises for elementary school'],
  'query_type': ['educational'],
  'timestamp': ['2023-09-20 14:30:00']},
 {'query': ['Don Quixote summary and analysis'],
  'query_type': ['literary'],
  'timestamp': ['2023-09-20 14:45:00']},
 ...
]

Current state of the user's Videos app:
[{'title': ['Analysis of Don Quixote by Miguel de Cervantes'],
  'duration_secs': tensor([3600]),
  'tags': [('Spanish literature',), ('Don Quixote',), ('Miguel de Cervantes',)],
  'category': ['Education'],
  'watch_timestamp': ['2023-09-20 20:30:00']},
 {'title': ['How to Play Flamenco Guitar for Beginners'],
  'duration_secs': tensor([2700]),
  'tags': [('Music',), ('Flamenco',), ('Guitar',)],
  'category': ['Music'],
  'watch_timestamp': ['2023-09-22 19:45:00']},
 ...
]

Current state of the user's Notes app:
[...,
 {'title': ['Paella Recipe'],
  'content': ['Ingredients: saffron, chicken, seafood, rice. Instructions: heat oil, add
  onion and garlic, add rice and cook until lightly toasted. Add broth and simmer until
  rice is cooked.'],
  'last_updated': ['2023-09-20 18:00']},
 {'title': ['Lesson Plan To-Do List'],
  'content': ['Prepare slides for Don Quixote, create reading comprehension questions,
  assign group project on character analysis. Due: 2023-10-12'],
  'last_updated': ['2023-10-05 10:00']},
 ...,
 {'title': ['Grocery Shopping List - Paella Ingredients'],
  'content': ['Saffron, chicken breast, shrimp, mussels, rice, onion, garlic, olive oil,
  seafood broth'],
  'last_updated': ['2023-10-03 12:00']}
]

Current state of the user's Maps app:
{'home_location': ['3456, Calle de la Luna, Madrid, Spain 28004'],
 'current_location': ['3456, Calle de la Luna, Madrid, Spain 28004']
}

Current state of the user's Music app:
{'currently_playing': ['']}
```

## Longitudinal Contexts Example

```
Here is a description of the user's current context: ["The video captures a person in a e
garage interacting with a bicycle, with the garage door open to reveal a driveway and
parked cars outside."]

Here are a list of the user's past actions with corresponding text context to help you
in this task:

{
"context": [
"In a bathroom, a person in a patterned robe uses a smartphone near a sink.
The scene includes a mirror reflecting the room and decorative plants. In a bathroom,
a person in a floral-patterned top interacts with a soap dispenser, holding it near the
sink. The scene includes a decorative plant and a towel, emphasizing a personal care
routine."
],
"action": [
"{'type': 'shop', 'content': 'hand soap'}"
],
"transcription": "Speaker 0: Do you\nSpeaker 0: know what we\nSpeaker 0: want?
Do you know what we want? Do you know what we want? Do you"
}

{
"context": [
"The video shows a person in a dimly lit room using a smartphone, possibly controlling
or interacting with a stereo system. The room contains various electronic devices and
furniture, with the person moving towards the stereo while holding the phone. The video
shows a person interacting with a gaming setup in an indoor space. The setup includes
a console or computer, controllers, and furniture such as a red couch or chairs. A fan
is present, and the room is moderately lit."
],
"action": [
"{'type': 'temporal_attention', 'action': 'set', 'time': '1:00:00', 'content':
'stop gaming'}"
],
"transcription": ""
}

{
"context": [
"In a nighttime setting, a person interacts with a smartphone near a parked blue car and
a bicycle, preparing to ride. The video captures a calm bicycle ride through a quiet
suburban area, highlighting the smooth movement and the serene environment with
minimal distractions."
],
"action": [
"{'type': 'control_environment', 'target': 'bike recording app', 'value': 'start'}"
],
"transcription": ""
}

{
"context": [
"In a bathroom, a person interacts with a tube near a sink and mirror.
The scene includes a white sink, a mirror, and a gray shirt with 'Windows.NET' text.
The person holds a tube with a red label. In a warmly lit bathroom, a person squeezes
a white and red tube. The scene includes a sink, mirror, and various bathroom items,
with the person wearing a gray T-shirt."
],
"action": [
"{'type': 'shop', 'content': 'toothpaste'}"
],
"transcription": ""
}
```

Next, we provide the prompt used to generate digital goals.

## Goal Generation Prompt

```
You are an intelligent agent.
Your task is to answer the following question:

Based on the images provided representing what the user is seeing,
what digital action might they want to do on their phones?
```

```
To answer the question above, you MUST PICK A DIGITAL ACTION from the following TEMPLATE:

{
  "type": "search",
  "source": ["world", "timeline"],
  // "world" = general knowledge (facts, news, weather, etc)
  // "timeline" = user's history or environment (e.g., saved events)
  "query_item": str,             // e.g., "Red delicious apple"
  "query": str                   // e.g., "Nutritional content"
}

{
  "type": "store_memory",
  "content": str                 // e.g., "I took my vitamins today"
}

{
  "type": "temporal_attention",
  "action": ["set", "unset"],
  "time": str,                   // e.g., "Ten minutes"
  "content": str                 // e.g., "Get ready for work"
}

{
  "type": "guided_activity",
  "content": str,                // e.g., "guided meditation"
  "time": str                    // e.g., "ten minutes"
}

{
  "type": "control_environment",
  "target": str,                 // e.g., "smart lights in kitchen"
  "value": str                   // e.g., "brightness: 50%"
}

{
  "type": "communication",
  "action": ["message", "share"],
  "target": str,                 // e.g., "Alex", "Instagram"
  "content": str
}

{
  "type": "translate",
  "modality": ["text", "audio"],
  "src_language": str            // e.g., "german"
}

{
  "type": "find_directions",
  "to": str                      // e.g., "nearest coffee shop"
}

{
  "type": "shop",
  "content": str                 // e.g., "running shoes"
}

{
  "type": "entertain",
  "modality": ["audio", "vision"],
  "content": str                 // e.g., "rock playlist", "cat videos"
}

Instructions:

- Choose only one action type that best fits the situation.
- Fill in all relevant fields in the template based on the visual context.
- DO NOT simply copy from the template. Write a meaningful, contextualized output.
```

We also present the prompt used by the LLM Judge to score generated goals.

### LLM Judge Prompt

```
You are a judge. Your task is to evaluate the digital action predicted by the model.
The digital action is a dictionary falling into one of the following goal types:
```

```
<SAME TEMPLATE AS IN GOAL GENERATION PROMPT>

Specifically, you are given a set of contexts that describe the video, as well as
the predicted digital action that the user might want to take on their phones given the
observed contexts.

You need to decide whether the predicted digital action is sensible given the contexts.
To facilitate this, you are further given a reference digital action that is known to be
sensible given the contexts.

Contexts:
{contexts}

Reference digital action:
{reference_digital_action}

Model predicted digital action:
{predicted_digital_action}

NOTE:

Please evaluate the prediction with a score that's either 0, 1 or 2 that corresponds to
"irrelevant", "borderline relevant", or "very relevant".

A score of 0 → The digital action would probably not be useful for the person in the video.

A score of 1 → The digital action might be useful for the person in the video,
but you're not confident.

A score of 2 → The digital action is definitely useful for the person in the video.

Please leverage the reference digital action when scoring the predicted digital action,
but do not treat it as the single possible answer due to the multi-modal nature of this
prediction task.

Please only output the score and nothing else. Do not add any explanation to your final
answer.

Wrap the score with <score> and </score> tags. For instance, an example full output
should look like this: <score>2</score>
```

Note that we used {0, 1, 2} judge scores in the prompt instead of {0, 0.5, 1.0} described in the paper, to use least amount of tokens (1 token rather than 3 comparing "1" vs "0.5"). We do post-processing to normalize them into {0, 0.5, 1.0}.

Finally, we provide the prompt used to generate video captions for longitudinal contexts. The prompt is adapted from [3].

**Longitudinal Contexts Generation Prompt**

```
MAIN INSTRUCTIONS:

Your task is to analyze video frames extracted from an Ego-centric video for a detailed
video understanding exercise.

Examine the video frames closely and generate a comprehensive caption by strictly
following the steps below:

Step 1: **Scene Context**:
Observe the video. What is the primary setting and activity in the video?

Step 2: **Spatial Relationship Analysis**:
Examine and report on the spatial relationships between key objects or characters in the
video frames. Describe the positioning and orientation of each element relative to others.

Step 3: **Detailed Object Analysis**:
List the key objects and characters in the frame. Describe their color, shape, texture,
and any other notable features with precision. Focus on specific details like clothing,
accessories, and colors.

Step 4: **Motion Description**:
Identify and describe any significant motion or actions taking place.

Step 5: **Text Analysis**:
Examine and return any significant texts observed in the video frames.
```

```
i.e.: menu, book title, billboards etc.

Step 6: **Additional Details**:
Note any other important details or elements that stand out but are not covered by the
above points, i.e.: gender, hair color, colors of accessories and other attributes in the
video frames.

Step 7: **Summary**:
Provide a concise yet comprehensive summary capturing the key elements and takeaways
from this video following Steps 1 to 6 above.
Your caption should encapsulate the scene's key aspects, offering a comprehensive
understanding of its environment, activities and context.

GUIDELINES:

1. Strictly return your results in JSON format. Please see the example below:
```json
{
  "Scene Context": "A busy beach scene with families and surfers enjoying the sunny day.",
  "Spatial Relationship Analysis": "The sandcastle is in the foreground, the dog
  approaches from the left, and the surfer moves from right to center.",
  "Detailed Object Analysis": "Children are wearing colorful swimwear; the dog is a golden
  retriever; the surfer is wearing a blue and white wetsuit.",
  "Motion Description": "Children are building a sandcastle, a dog is running towards
  the water, and a surfer is catching a wave.",
  "Text Analysis": "There is a billboard on the beach with the ads that reads as Diving
  Equipment Rental",
  "Additional Details": "One child has red hair; the dog's leash is lying abandoned on
  the sand; multiple surfboards are visible in the background.",
  "Summary": "The video frames depict a joyful beach day emphasizing family activities,
  interaction with nature, and surfing as a key activity, showcasing the beach's vibrant
  atmosphere."
}
2. When not sure, please be conservative and restrain from adding uncertain information
to your response. Do not make up facts. Do not make up information.
3. VERY IMPORTANT: YOU ARE ALLOWED TO USE A MAXIMUM OF 200 words in total.
```

