# OpenReview forum: "Benchmarking Egocentric Multimodal Goal Inference for Assistive Wearable Agents"
_NeurIPS.cc/2025/Datasets_and_Benchmarks_Track — NeurIPS 2025 Datasets and Benchmarks Track spotlight_

### Official Review · Reviewer_V6Ef · 2025-06-18

**Rating:** 5
**Confidence:** 4

**Summary:**

The paper introduces the first benchmark that targets automatic inference of a user’s high-level goal from egocentric multimodal observations obtained through wearable devices.
- Dataset: 3 477 scripted recordings (~29 h) from 348 participants captured with Meta Aria smart-glasses. Each sample provides four synchronized modalities: vision, audio, synthetic digital context (seven app states) and longitudinal history (textual Socratic summaries of earlier clips).
- Benchmark tasks: (i) discriminative multiple-choice (MCQ) goal selection; (ii) open-ended generation scored by an LLM “judge”.
- Human study establishes an upper bound (≈93 % MCQ accuracy); the best vision-language model (VLM) reaches 84 % (MCQ) and ~0.55 on the generative LLM-judge metric.
- Extensive ablations quantify the contribution of each modality and show that large models cope better with noisy digital / longitudinal signals while smaller, edge-suitable models lag far behind.
- Meta-evaluation demonstrates that an LLM judge conditioned on reference goal + script cues agrees with human raters (76.8 % vs 75.2 % human-human).

**Dataset Code Accessibility:**

Yes

**Ethical Considerations:**

No, there are no or only very minor ethics concerns

**Final Justification:**

All my concerns have been fully addressed in the authors' rebuttal. I am satisfied with their responses and recommend acceptance.

**Limitations Weaknesses:**

1. Digital and longitudinal contexts are LLM-generated rather than captured from real user devices, raising a domain-gap risk when models are deployed on actual wearables. Ablation results already show low signal-to-noise; the synthetic nature may exaggerate this effect.
2. Although scripts increase annotation fidelity, they limit spontaneity; users rarely act exactly according to a script in the wild. The benchmark may over-estimate achievable performance in truly unconstrained settings.
3. Videos average 30 s and histories consist of only five prior clips. Many real scenarios (e.g., “Where did I park yesterday?”) require hours- or days-long context.
4. Benchmarking a single model requires multi-modal input concatenation (especially digital + longitudinal text) that exceeds context windows of many open-source VLMs. Guidance on pruning or chunking would help the community.

**Strengths Contributions:**

1. Proactive goal inference is a natural next step for wearable assistants; framing it as a multimodal language problem is timely and well motivated.
2. First to combine egocentric video, audio, digital app state and longitudinal traces with ground-truth goals. Scripted data collection allows precise control over which modality contains the discriminative cues; this enables targeted ablations absent in prior work (Ego4D, EgoLife, MM-Ego). Real human participants across diverse environments improve ecological validity over entirely synthetic datasets.
3. Detailed data statistics, annotation protocols, IRB approval, human-subject instructions, compute budgets and check-list compliance are exemplary for a D&B submission.

---

> ### Author Rebuttal · Authors · 2025-07-31
>
> # Author response to Reviewer V6Ef
> We thank you for your careful review of our submission. Thank you for acknowledging the importance of our research on proactive goal inference framed as a multimodal language problem. We appreciate your recognition of our benchmark as the first effort to combine synchronized egocentric video, audio, digital app state and longitudinal modalities for goal inference with scripted data collection. We also appreciate your thorough review of our data statistics, annotation protocols and behavioral experiment details we mention in the paper.
>
> Please find our response to each of your critiques below:
>
> ### 1. Digital and longitudinal contexts are LLM-generated rather than captured from real user devices
>
> This is a fair point, and while we acknowledge the potential domain shifts introduced by using LLM-generated digital and longitudinal contexts, our approach prioritizes user privacy and ensures a diverse, unbiased benchmark for evaluating goal inference in wearable devices.
>
> Collecting real user digital states entails capturing the full device state containing private information (device memory such as personal photos and audios, app states with private information such as user’s location, messages, contacts, etc.) as part of the benchmark. This would compromise our aim to contribute a benchmark that the community as a whole can use to evaluate progress on goal inference.
>
> We note that our longitudinal context is not completely LLM generated, but rather socratic context is generated from videos corresponding to our longitudinal scenarios. This way, our benchmark can be used to test whether current VLMs are capable of integrating memory of the user's past interactions with the system to make personalized goal predictions. We don’t rely on mining for naturally occurring longitudinal relationships from an unconstrained data collection paradigm, which could result in sparse and biased longitudinal interactions which would allude to a more noisy/limited estimation of accuracy on longitudinal goal inference.
>
> Additionally, we highlight the inherent tradeoff between ecological validity of real user digital states / longitudinal contexts and the diversity of collected data. Our benchmark covers a broad range of app interactions that may not fully be covered in real user data (biased by popularity of some apps and interactions over others). Likewise, deliberately scripting longitudinal scenarios allows us to robustly benchmark longitudinal goal inference performance of VLMs under many diverse ecologically valid wearable assistant use cases.
>
> Finally, we would like to highlight that state-of-the-art LLMs are excellent and being widely used at synthetic data generation [1, 2, 3].
>
> ### 2. Although scripts increase annotation fidelity, they limit spontaneity
> We acknowledge that scripting data collections may limit spontaneity, but we see this as a feature rather than a bug. Our goal with this proposed benchmark is to create the first large-scale multimodal proactive goal inference benchmark for wearable assistants. To this end, we prioritized (1) high fidelity of annotations, (2) increased diversity of scenarios, and (3) precise control over ablation of different modalities to identify their relative importance during goal inference.
>
> Scripted data collection allows us to ensure the above three criteria. We believe that at the current state of evaluating VLMs on goal inference, a scripted high quality benchmark such as ours with diverse scenarios and robust annotations is relatively more important. We believe our findings from this benchmark inspire further exploration of model performance in more spontaneous, unconstrained proactive goal inference settings.
>
> ### 3. Videos average 30 s and histories consist of only five prior clips. Many real scenarios (e.g., “Where did I park yesterday?”) require hours- or days-long context.
> We observe a trade-off between collecting longer-horizon videos and including a large number of varied scenarios at scale with real participants. Prior work, such as EgoLife [4], which covers longer-horizon tasks, often involves a limited number of participants. The cost of collecting diverse scenarios outweighs the benefits of gathering data within the same environment over extended periods.
>
> ### 4. Benchmarking a single model requires multi-modal input concatenation (especially digital + longitudinal text) that exceeds context windows of many open-source VLMs. Guidance on pruning or chunking would help the community.
> We agree that context window requirements of multimodal inputs is a valid problem. We will add a discussion point on how one could also use smaller models with pruning or chunking.
>
> Subsequently, our results also show that larger models with long context sizes generally work better on egocentric goal inference, and this finding inspires further engineering efforts from the community such as distillation / quantization to make large performant VLMs more efficient.
>
> References:
> 1. Nadas, M., Diosan, L., & Tomescu, A. (2025). Synthetic data generation using large language models: Advances in text and code. arXiv preprint arXiv:2503.14023.
> 2. Wu, S., Xu, Z., Zhang, Y., Zhang, Y., & Ramage, D. (2024). Prompt public large language models to synthesize data for private on-device applications. arXiv preprint arXiv:2404.04360.
> 3. Long, L., Wang, R., Xiao, R., Zhao, J., Ding, X., Chen, G., & Wang, H. (2024). On llms-driven synthetic data generation, curation, and evaluation: A survey. arXiv preprint arXiv:2406.15126.
> 4. Yang, J., Liu, S., Guo, H., Dong, Y., Zhang, X., Zhang, S., ... & Liu, Z. (2025). Egolife: Towards egocentric life assistant. In Proceedings of the Computer Vision and Pattern Recognition Conference (pp. 28885-28900).

---

> > ### Comment · Reviewer_V6Ef · 2025-08-04
> >
> > Thank you to the authors for your detailed and thoughtful response to my review. I have read your rebuttal carefully and appreciate the effort and consideration you have put into addressing my concerns.
> > Overall, your response clearly and convincingly addresses all of my concerns.

---

> > > ### Author Response · Authors · 2025-08-07
> > > **Author Response to Comment by Reviewer V6Ef**
> > >
> > > Dear Reviewer,
> > >
> > > We are glad that our rebuttal convincingly addresses your concerns. We thank you for reading our response to your review and for engaging in the author-reviewer discussion.

---

### Official Review · Reviewer_bpsq · 2025-07-02

**Ethics Flags:** Data privacy, copyright, and consent
**Rating:** 5
**Confidence:** 4

**Summary:**

This paper introduces a benchmark for egocentric multimodal goal inference in assistive wearable agents. The dataset contains vision, audio, digital, and longitudinal data with ground truth goal. They propose two evaluation paradigms, a discriminative multiple‐choice task with human and model baselines and an open‐set generative task scored by an LLM judge. The evaluation shows there stills has gaps between VLMs and human in predicting the goal. On the generation task, the LLM judge with a reference goal or script cues can effectively substitute human evaluators.

**Additional Feedback:**

Fairness is a critical aspect of this benchmark. Introducing an LLM judge during inference could compromise that fairness. The authors should include measures to improve the benchmark’s generalizability.

**Dataset Code Accessibility:**

Yes

**Dataset Code Comments:**

The code structure is clear and easy to access.

**Ethical Comments:**

The authors do not specify whether and how they obtained user consent for the data.

**Ethical Considerations:**

Yes, there are ethics concerns that require attention by the authors

**Final Justification:**

I understand the authors’ explanation regarding the data issue, and I think using synthetic data is a reasonable and practical solution in this context. This addresses my main concern.

**Limitations Weaknesses:**

1. The digital contexts are entirely LLM-synthesized rather than collected from real users, which may fail to reflect genuine user behavior.
2. Human-involved evaluation is limited, potentially impacting the dataset’s generalizability. Although the meta-evaluation supports the LLM judge, the core goal-inference task is designed for humans rather than LLMs, which could introduce biases when extending to automated inference.
3. The dataset is heavily biased toward indoor environments, limiting its applicability to outdoor or mixed settings.
4. The benchmark does not take device response latency into account, which is a crucial factor in user experience.

**Strengths Contributions:**

1. This work introduces the first dataset for wearable-agent goal inference, a timely and impactful contribution to the advancement of wearable-agent research.
2. The dataset spans different user real-world scenarios and includes two evaluation protocols for comprehensive assessment. It comprises 29 hours of multimodal recordings from 348 participants over 3,477 sessions.
3. A comprehensive ablation study quantifies each modality’s contribution to inference performance, revealing that small and medium models suffer performance degradation when integrating multiple modalities.
4. The LLM-based judge reliably detects semantic similarities.

---

> ### Author Rebuttal · Authors · 2025-07-31
>
> # Author response to Reviewer bpsq
>
> We thank you for your careful review of our submission. We appreciate your recognition of our work introducing the first dataset for wearable-agent goal inference and its impact for advancing research in this area. Thank you also for recognizing the comprehensive nature and scale of our multimodal dataset and for your interest in the ablation of each modality’s contribution to goal inference performance across models of varying sizes.
>
> Please find our response to each of your critiques below:
> ### 1. “The digital contexts are entirely LLM-synthesized rather than collected from real users, which may fail to reflect genuine user behavior.”
> This is a fair point, and while we acknowledge the potential domain shifts introduced by using LLM-generated digital and longitudinal contexts, our approach prioritizes user privacy and ensures a diverse, unbiased benchmark for evaluating goal inference in wearable devices.
>
> Collecting real user digital states entails capturing the full device state containing private information (device memory such as personal photos and audios, app states with private information such as user’s location, messages, contacts, etc.) as part of the benchmark. This would compromise our aim to contribute a benchmark that the community as a whole can use to evaluate progress on goal inference.
>
> Additionally, we highlight the inherent tradeoff between ecological validity of real user digital states / longitudinal contexts and the diversity of collected data. Our benchmark covers a broad range of app interactions that may not fully be covered in real user data (biased by popularity of some apps and interactions over others). Likewise, deliberately scripting longitudinal scenarios allows us to robustly benchmark longitudinal goal inference performance of VLMs under many diverse ecologically valid wearable assistant use cases.
>
> Finally, we would like to highlight that state-of-the-art LLMs are excellent and being widely used for synthetic data generation [1, 2, 3].
>
>
> ### 2. “Ethical comments: The authors do not specify whether and how they obtained user consent for the data.”
> Thank you for careful consideration of user consent for the data used in our benchmark. ALL of our data was scripted and collected with paid third party vendors who gave their consent to take part in our data collection. Any PII information of the data collectors has been redacted from dataset files. We will include a copy of the user consent form we used as part of our data collection in the main release and add a user consent statement to the paper.
>
>
> ### 3. “Human-involved evaluation is limited, potentially impacting the dataset’s generalizability. Although the meta-evaluation supports the LLM judge, the core goal-inference task is designed for humans rather than LLMs, which could introduce biases when extending to automated inference.”
> We respectfully disagree with the above critique. We don’t agree with the reviewer’s assumption that LLM judgement is misaligned with human judgment of goal inference performance. As the reviewer notes, our meta-evaluation shows that an LLM judge with access to a reference goal and script has higher alignment to humans than inter-annotator agreement.
>
> We would like to highlight that human-involved evaluation at scale is very expensive and intractable as the benchmark grows in size. Given our meta-evaluation results supporting the LLM Judge, we find this a reasonable way to gauge performance on open-ended goal inference that is closely aligned with human judgment. Prior work on open ended generative evaluation using LLMs supports our position [4,5].
>
> We refer the reviewer to Sections B.2 and C of our Appendix for more information on human-involved evaluation with example screenshots. In addition, we will release the set of MCQ and generative instances we used in our human experiments along with corresponding videos and trial-by-trial responses needed for reproducibility of our results.
>
>
> ### 4. “Fairness is a critical aspect of this benchmark. Introducing an LLM judge during inference could compromise that fairness. The authors should include measures to improve the benchmark’s generalizability.”
> All models are assessed using the same LLM Judge, which is compared to human evaluations through a carefully designed behavioral experiment. Since all models are evaluated uniformly, the argument that the LLM Judge compromises fairness lacks validity. Our meta-evaluation results also demonstrate a high similarity between human and LLM judgments, supporting the use of the LLM Judge as a reliable metric for benchmarking models in open-ended generative goal inference.
>
> ### 5. “The dataset is heavily biased toward indoor environments, limiting its applicability to outdoor or mixed settings.”
> We agree that our benchmark contains a limited number of outdoor scenarios. However, we note that (as evidenced by our results) goal inference in indoor environments is still an unsolved, challenging problem. As mentioned in your review, our benchmark contributes the first large-scale multimodal benchmark to measure progress on this challenging problem. We will emphasize clearly in the paper that our benchmark comprises a larger proportion of indoor scenarios.
> ### 6. “The benchmark does not take device response latency into account, which is a crucial factor in user experience.”
> Our benchmark’s aim is the measurement of goal inference performance for wearable assistants, and we evaluate models with varying sizes (and latency) and competence. Our contribution is an objective measurement of goal inference performance of a comprehensive suite of state-of-the-art VLMs. We leave it to future work for establishing goal inference performance as a function of model latency; this is out of the scope of our submission.
>
> References
> 1. Nadas, M., Diosan, L., & Tomescu, A. (2025). Synthetic data generation using large language models: Advances in text and code. arXiv preprint arXiv:2503.14023.
> 2. Wu, S., Xu, Z., Zhang, Y., Zhang, Y., & Ramage, D. (2024). Prompt public large language models to synthesize data for private on-device applications. arXiv preprint arXiv:2404.04360.
> 3. Long, L., Wang, R., Xiao, R., Zhao, J., Ding, X., Chen, G., & Wang, H. (2024). On llms-driven synthetic data generation, curation, and evaluation: A survey. arXiv preprint arXiv:2406.15126.
> 4. Li, H., Dong, Q., Chen, J., Su, H., Zhou, Y., Ai, Q., ... & Liu, Y. (2024). Llms-as-judges: a comprehensive survey on llm-based evaluation methods. arXiv preprint arXiv:2412.05579.
> 5. Xiong, T., Wang, X., Guo, D., Ye, Q., Fan, H., Gu, Q., ... & Li, C. (2025). Llava-critic: Learning to evaluate multimodal models. In Proceedings of the Computer Vision and Pattern Recognition Conference (pp. 13618-13628).
>
>
> ### Summary of changes and deliverables to be made available with public release:
> 1. User consent statement and link to user consent form we used for our data collection
> 2. MCQ and generative instances from the human experiment, along with videos and trial-by-trial responses.
> 3. Emphasize in paper that our dataset comprises a larger proportion of indoor scenarios.

---

> > ### Comment · Reviewer_bpsq · 2025-08-03
> >
> > Thank you for the detailed response. It addressed several of my concerns and clarified important aspects of the work. I appreciate the authors’ efforts to improve the dataset and evaluation methodology. I still have following question.
> > I agree that photos can be highly sensitive and may not be suitable for direct release. However, face blurring and similar anonymization techniques are commonly used in practice. Additionally, if the photos are also generated by LLMs, I am uncertain whether such synthetic visual data is appropriate for this task.

---

> > > ### Author Response · Authors · 2025-08-07
> > > **Author response to Official Comment by Reviewer bpsq**
> > >
> > > Dear Reviewer,
> > >
> > > We thank you for reading our rebuttal and for engaging in the author-reviewer discussion. We are glad that our response addressed several of your concerns and clarified important aspects of our work. We respond below to your follow-up question:
> > >
> > > - To clarify, we did not use generative AI to produce images; all digital states are represented in the form of text.
> > > - In addition, all video data was captured in consented environments with all participants agreeing to do data collection prior to and recordings being captured. Please see our response to the ethics review for more information.
> > > - In case you are asking about releasing real device states from participant phones – a hypothetical approach we mentioned in the rebuttal – this would be inappropriate even with proper anonymization (which would be extremely difficult given the amount of personally-identifiable-information hidden on a phone): real user phone state is not generated with respect to a scripted scenario. Therefore, it would be impossible to ensure that the phone state contains the relevant information (for example, ensuring that the user has a recent text message asking about whether the dog has been fed in the top example from Figure 1 in the paper).

---

### Official Review · Reviewer_Xmid · 2025-07-03

**Rating:** 5
**Confidence:** 3

**Summary:**

This paper presents a benchmark for egocentric multimodal goal inference for assistive wearable agents. It is a well motivated task to reduce the friction of using wearable agents by tasking the agents to guess the user's intention and ideally automatically execute for the user. At first glance, it is not a easy to benchmark such task since it is difficult to curate such dataset and design evaluation metrics. To facilitate the benchmark, they collect a new dataset with 29h of multimodal data from 348 participants across 3477 recordings, each annotated with goal or intentions. The dataset has rich modalities, including video, audio, digital contexts and longitudinal contexts. They carefully design discriminative (MCQs) and generative benchmarks. Beyond the dataset curation, this paper also presents thorough study on the evaluation metrics. They study the alignment between human raters and LLM-as-Judge to justify their usage of LLMs as the judge for the generative benchmark. With such well designed benchmark, they tested multiple MLLMs, including open-source and closed-source ones and reveal the limitations of current MLLMs in egocentric multimodal goal inference.

**Additional Feedback:**

NA

**Dataset Code Accessibility:**

Yes

**Dataset Code Comments:**

Link to the dataset and relavent codes are available.

**Ethical Comments:**

Egocentric video audio capture is broadly performed by many previous works. Digital and longitudinal contexts are generated by LLMs. Therefore, I do not see any ethical concerns.

**Ethical Considerations:**

No, there are no or only very minor ethics concerns

**Final Justification:**

I thank authors for the rebuttal. I read through it and all my concerns and questions are addressed. Authors admit some limitations of the proposed benchmark, and identify a few potential future work. I therefore keep my original score of accept.

**Limitations Weaknesses:**

- The digital contexts are randomly generated by LLMs, which may not be a good representation of real life digital contexts.
- The benchmark only contains a fix set of 165 scripts generated by LLMs, which may not achieve a good coverage of real use cases.
- The diversity of recording locations is limited, with most of the recordings happening indoor.
- It is definitely interesting to see what would happen if you go one more step further, executing the inferenced goals for user if applicable and see how do users like the result.
- It is also interesting to test on LLMs that can consume multiple different modalities like video, audio, app screenshots natively, and compare if using these modalities natively are better than using socratic contexts/ audio transcipts.

**Strengths Contributions:**

+ The proposed dataset is the first egocentric multimodal dataset that includes digital contexts, which is definitely a useful and important modality for the future study of wearable agents.
+ The proposed benchmark is the first one that incorporates video, audio, digital and longitudinal contexts for egocentric goal inference.
+ The curation process of the benchmark, including scripts generation, audiovisual context generation, digital context generation, longitudinal context generation, and tasks generation, are carefully designed, executed and validated.
+ The meta-evaluation that proves the effectiveness of LLM-as-Judge is interesting and important to not only this benchmark, but also many other ones using LLM-as-Judge.
+ The benchmark reveals several limitations of current MLLMs. Overall, the strongest MLLMs now only achieves accuracy of 55%, and their model size is not suitable for wearable / edge devices. They also struggle to use information with low signal-to-noise ratio.

---

> ### Author Rebuttal · Authors · 2025-07-31
>
> # Author response to Reviewer Xmid
> We thank you for your careful review of our submission. Thank you for recognizing our work introducing the first egocentric multimodal dataset that includes digital and longitudinal context, both important modalities for future study of wearable agents. We appreciate your recognition of our carefully designed and executed dataset generation and evaluation. We also recognize our meta-evaluation results as applicable for many other benchmarks using LLM-as-a-Judge. It is indeed interesting that even the strongest, very large MLLMs still only achieve 55% accuracy on open-ended goal inference.
>
> Please find our response to each of your critiques below:
> ### 1. “The digital contexts are randomly generated by LLMs, which may not be a good representation of real life digital contexts.”
> This is a fair point, and while we acknowledge the potential domain shifts introduced by using LLM-generated digital and longitudinal contexts, our approach prioritizes user privacy and ensures a diverse, unbiased benchmark for evaluating goal inference in wearable devices.
>
> Collecting real user digital states entails capturing the full device state containing private information (device memory such as personal photos and audios, app states with private information such as user’s location, messages, contacts, etc.) as part of the benchmark. This would compromise our aim to contribute a benchmark that the community as a whole can use to evaluate progress on goal inference.
>
> Additionally, we highlight the inherent tradeoff between ecological validity of real user digital states / longitudinal contexts and the diversity of collected data. Our benchmark covers a broad range of app interactions that may not fully be covered in real user data (biased by popularity of some apps and interactions over others). Likewise, deliberately scripting longitudinal scenarios allows us to robustly benchmark longitudinal goal inference performance of VLMs under many diverse ecologically valid wearable assistant use cases.
>
> Finally, we would like to highlight that state-of-the-art LLMs are excellent and being widely used for synthetic data generation [1, 2, 3].
>
> ### 2. “The diversity of recording locations is limited, with most of the recordings happening indoor.”
> We agree that our benchmark contains a limited number of outdoor scenarios. However, we note that (as evidenced by our results) goal inference in indoor environments is still an unsolved, challenging problem. We will emphasize clearly in the paper that our benchmark comprises a larger proportion of indoor scenarios.
>
> ### 3. “The benchmark only contains a fix set of 165 scripts generated by LLMs, which may not achieve a good coverage of real use cases.”
> Our set of 165 scripts was selected with coverage of diverse real-life use cases of wearable assistants covered. Although LLMs were used as a tool for some script generation, each script was carefully selected with aggressive rejection for emphasizing diversity (e.g. redundant scenarios filtered out), hand-verified and often rewritten / vetted by multiple human experts for resulting quality and diversity.
>
> ### 4. “It is also interesting to test on LLMs that can consume multiple different modalities like video, audio, app screenshots natively, and compare if using these modalities natively are better than using socratic contexts/ audio transcipts.”
> We chose a widely applicable approach by using socratic modeling to estimate performance, as many models utilize language context. While we acknowledge that native multimodal performance might differ, it is also compute-intensive. We anticipate that future work will explore this direction, and our work provides a metric for measuring such performance using natively multimodal models.
>
> ### 5. “It is definitely interesting to see what would happen if you go one more step further, executing the inferenced goals for user if applicable and see how do users like the result.”
> Thank you for your interest in evaluating user feedback on executing the inferred goals. We believe our generative LLM-Judge evaluation sets the stage for measuring user feedback in a closed loop system with execution of inferred goals. We anticipate future work to explore this direction.
>
> References:
> 1. Nadas, M., Diosan, L., & Tomescu, A. (2025). Synthetic data generation using large language models: Advances in text and code. arXiv preprint arXiv:2503.14023.
> 2. Wu, S., Xu, Z., Zhang, Y., Zhang, Y., & Ramage, D. (2024). Prompt public large language models to synthesize data for private on-device applications. arXiv preprint arXiv:2404.04360.
> 3. Long, L., Wang, R., Xiao, R., Zhao, J., Ding, X., Chen, G., & Wang, H. (2024). On llms-driven synthetic data generation, curation, and evaluation: A survey. arXiv preprint arXiv:2406.15126.

---

### Official Review · Reviewer_9C7y · 2025-07-04

**Rating:** 5
**Confidence:** 2

**Summary:**

This paper introduces a comprehensive benchmark for evaluating goal inference capabilities in assistive wearable agents. A dataset comprising 29 hours of multimodal data from 348 participants across 3,477 recordings is collected, featuring ground-truth goals alongside visual, audio, digital, and longitudinal contextual observations. The benchmark evaluates vision-language models (VLMs), with human performance establishing an upper bound at 93% accuracy compared to 84% for the best VLM.

**Dataset Code Accessibility:**

Yes

**Dataset Code Comments:**

The authors released the dataset and the code on two public web links, respectively, which are accessible.

**Ethical Considerations:**

No, there are no or only very minor ethics concerns

**Final Justification:**

I have read the response to my comments as well as those from other reviewers. I think that the main value of this paper is proposing the first dataset on egocentric goal inference. Although this dataset is synthetic, it still provides a new sight on this direction. On the other hand, I do suggest the authors to clarify the possible domain gap between the synthetic data and the real scene, and discuss how to narrow such gap in the future work.

**Limitations Weaknesses:**

Since the dataset addresses a new problem for wearable agent use, it would be helpful to discuss deployment constraints for wearable devices, and how these constraints may affect real-world applicability.

**Strengths Contributions:**

1. The paper introduces the first large-scale dataset specifically designed for egocentric goal inference. The dataset is comprehensive, incorporating four modalities (vision, audio, digital, and longitudinal) with 348 diverse participants, covering realistic wearable agent use cases.
2. The dataset is carefully designed. The 165 scripted scenarios are specifically selected to ensure multimodal relevance; the three-rater annotation system with acceptance criteria ensures high-quality ground truth.
3. The paper provides a thorough evaluation framework that includes both discriminative (MCQ) and generative (LLM-as-judge) evaluation methods, evaluating multiple model families (Llama, Qwen2.5-VL, InternVL-2.5) across different scales, and makes comparison against human judgment.

---

> ### Author Rebuttal · Authors · 2025-07-31
>
> # Author response to Reviewer 9C7y
>
> We thank you for your careful review of our paper. We appreciate your acknowledgement of our paper introducing the first large-scale benchmark for egocentric goal inference comprising four modalities (vision, audio, digital, and longitudinal) covering realistic wearable agent use cases. We thank you for recognizing the careful design and execution of our benchmarking, the confidence in our multi-rater annotation system and in our comprehensive evaluation of several model families at varying sizes using both discriminative and generative evaluation methods.
>
> Please find our response below to your critique of our submission:
> ### 1. "Since the dataset addresses a new problem for wearable agent use, it would be helpful to discuss deployment constraints for wearable devices, and how these constraints may affect real-world applicability."
> This is a valid point and relevant to Reviewer V6EF’s comment on pruning or chunking for open source VLMs. Our benchmark’s aim is the measurement of goal inference performance for wearable assistants, and we evaluate models with varying sizes (and latency) and competence. But we agree that it would be interesting to study the deployment constraints of wearable assistants. Our contribution is an objective measurement of goal inference performance of a comprehensive suite of state-of-the-art VLMs. Future work shall study goal inference performance as a function of model latency and deployment constraints; this is out of the scope of our current submission.

---

> > ### Comment · Reviewer_9C7y · 2025-08-04
> >
> > I have read the response to my comments as well as those from other reviewers. I think that the main value of this paper is proposing the first dataset on egocentric goal inference. Although this dataset is synthetic, it still provides a new sight on this direction. On the other hand, I do suggest the authors to clarify the possible domain gap between the synthetic data and the real scene, and discuss how to narrow such gap in the future work.

---

> > ### Author Response · Authors · 2025-08-07
> > **Response to Official Comment by Reviewer 9C7y**
> >
> > Dear Reviewer,
> >
> > We thank you for engaging in the author-reviewer discussion process. We appreciate your positive feedback and re-iterating the value of our paper in proposing the first egocentric goal inference dataset. Thanks for your attention to the issue on the possible domain gap between synthetic and real data (digital and socratic longitudinal context). We will clarify the possible domain gap, and clearly emphasize the domain gap - user privacy tradeoff (that we discussed in our responses to Reviewers Xmid, bpsq and V6ef) in our paper. We shall also include guidance for future work on addressing the above domain gap by improving LLMs to generate more realistic digital states and developing privacy-preserving methods for scaling digital state and longitudinal data collection.

---

### Note · Authors · 2025-08-15

We find it encouraging that all reviewers unanimously recognize the high novelty, impact, and rigor of our proposed egocentric goal inference benchmark. We addressed the main concerns raised by our reviewers as we discuss below and commit to comply with NeurIPS ethics guidelines.

## Strengths Recognized by Reviewers
1. Novelty and Impact: First large-scale benchmark for egocentric goal inference in wearable agents, combining four modalities (vision, audio, digital, longitudinal). Scale, diversity, and design are significant contributions.
2. Comprehensive Evaluation: Thorough evaluation across model families, sizes, discriminative and generative metrics, ablations, and modality insights.
3. Annotation Quality and Meta-Evaluation: Multi-rater annotation and LLM-as-a-Judge evaluation are robust and align well with human judgment.
4. Reproducibility and Transparency: Commitment to releasing data, protocols, evaluation scripts, and attention to user consent and privacy.

## Main Weaknesses and Our Responses
1. Synthetic data generation: We prioritized user privacy and benchmark diversity; real user data collection would compromise privacy and limit use. Synthetic data enables broad scenario coverage and aligns with recent trends. We will clarify this trade-off and discuss potential biases in the paper.
2. Dataset biased toward indoor environments: We will emphasize this limitation. Indoor goal inference still remains challenging as highlighted by our findings and serves as a foundation for future outdoor benchmarking.
3. Scripted data may limit spontaneity: Scripted collection ensures annotation fidelity, scenario diversity, and controlled ablations, appropriate for a first benchmark.
4. LLM-based evaluation may limit generalizability: Meta-evaluation shows LLM judgments align better with humans than inter-annotator agreement. Uniform LLM-Judge evaluation for all models ensures fairness. Human evaluation data and protocols will be released.

## Ethics Review: Paper changes
1. Adding subsections on human subject protocols, informed consent, privacy and anonymization, participant demographics and dataset biases, and environmental sustainability.
2. Summary of benchmarking compute cost and carbon footprint will be provided with recommendations for efficient evaluation.

We thank the reviewers and Area Chair for their consideration. We believe our benchmark provides a valuable foundation for future research in egocentric goal inference and wearable agents.

---

### Decision · Program_Chairs · 2025-09-18

**Decision:**

Accept (spotlight)

**Comment:**

The paper received four reviews and all reviewers voted for “accept“. This is the first large-scale egocentric goal inference benchmark designed specifically for wearable assistant agents. The evaluation is comprehensive, which covers a wide range of model families, sizes, and metrics. The main concerns are that the digital contexts are entirely synthesized using LLMs and the recording environments are mostly limited to indoors. During the rebuttal, the authors explained that using LLMs to generate digital contexts can largely protect user privacy, which makes sense. Regarding the limited diversity  of the recording environments, the authors acknowledged the issue but pointed out that indoor goal inference remains as a very challenging problem, which sounds reasonable. Since all reviewers unanimously voted for acceptance and the merits are clear and strong, the AC recommends that the paper be accepted. Recently, wearable assistant agents are gaining huge attention due to the increase in popularity of wearable devices like AI glasses. This work presents a timely benchmark that can facilitate future research on egocentric AI agents.